# Trace element systematics constrain the origin of fluids that form gem-quality diamonds
Aleksandr Rakipov [1,2,3] ✉, Alan B. Woodland[1,2], Fabrizio Nestola [3], Matilde Galiè[1,3], Martha G. Pamato [3], Davide Novella [3], Maxwell C. Day [3], Tobias Erhardt[1,2] & Wolfgang Müller [1,2]

Diamonds crystallise from fluids/melts circulating in the Earth's mantle. Analysis of these fluids is possible if they remain entrapped in the diamond during growth, but such fluid inclusions are rarely observed in gem-quality stones. We investigated thin films surrounding mineral inclusions, previously described as silicic fluid rims containing $Si_2O(OH)_6$ and $Si(OH)_4$, in gem-quality lithospheric diamonds from Siberia. Using micro-Raman spectroscopy and **L**aser-**A**blation **I**nductively-**C**oupled-**P**lasma-**M**ass-**S**pectrometry (LA-ICPMS) depth-profiling, we obtained compositional data from silicic fluid rims surrounding both silicate and non-silicate inclusions. Slow LA-ICPMS depth-profiling at the diamond-inclusion interface enabled differentiating the respective diamond/fluid-rim/mineral-inclusion contributions, allowing detection of Sr, Nb, Ba, La, Ce, Nd, and Th from the fluid rim. Here, we compare silicic fluid rims and other mantle-derived fluids/melts based on trace element ratios relative to the primitive mantle. Their $(La/Nb)_N$_$(Ba/Nb)_N$ and $(Nd/Nb)_N$_$(Th/Nb)_N$ systematics align with primitive mantle-like high-density fluids and group 2 kimberlites, suggesting an origin from kimberlite-like melts.

Lithospheric diamonds grow from metasomatic fluids/melts that percolate through the mantle and chemically interact with surrounding rocks[1–3]. Relics of these fluids may be preserved as inclusions in diamonds, which, together with coexisting mineral inclusions, provide direct evidence of mantle compositions, diamond-forming reactions, and deep volatile cycling[2,4–6].

The most extensively studied diamond-hosted fluids are high-density fluids (HDFs) occurring in *fibrous* diamond. Although HDFs are often interpreted to represent entrapped melts, they are referred to as fluids, following the definition that all mobile phases under entrapment conditions are considered fluids[1]. Here, we use the term *fibrous* to refer collectively to non-gem, fluid-rich regions of coated, cloudy, and cuboid or fully fibrous diamonds[1]. Based on major-element compositional analyses of HDFs, four distinct fluid groups are recognised: saline, high-Mg carbonatitic (HMC), low-Mg carbonatitic (LMC) and silicic[1]. Their trace-element patterns define two endmembers: *planed*, which are linked to asthenospheric sources, and *ribbed*, interpreted as indicating interaction with the subcontinental lithospheric mantle[7–9].

Similar compositions have also been documented in gem-quality diamonds, where major element data from fluid inclusions trapped along twin planes[10] and trace-element data from *invisible* nano-inclusions[5] closely match the major and trace element composition of HDFs, suggesting similar fluid sources for the formation of *fibrous* and gem diamonds. Together with a recent study that investigated the $CO_2$ content of fluids in healed diamond fractures[11], such investigations remain scarce because these fluids are optically invisible and difficult to locate in gem diamonds.

A second type of fluid observed in gem-quality diamonds, described as silicic fluid rims (SFRs)[12], occurs as a thin (<1–5 μm) film around minerals or as isolated inclusions[13]. Its ubiquitous occurrence and close association with mineral inclusions make it a relatively accessible target. The Raman spectrum of SFRs exhibits broad doublet in the 660–800 $cm^{-1}$ range, previously attributed to $Si_2O(OH)_6$ and $Si(OH)_4$ based on experimental studies on silica speciation in hydrous fluids[14,15]. Although isolated silicic fluid inclusions have also been observed along twin planes in macle diamonds[13], the major and trace element composition of these fluid rims has not been determined, and thus their chemical composition remains unknown.

The SFRs and HDFs have different physical properties. While HDF inclusions are often composed of daughter phases that crystallised after entrapment[1], SFRs display low-density features in X-ray tomography (SXRTM) and lack secondary crystallisation features[12]. These differences may reflect either a distinct fluid type (composition) or different stages of differentiation of the diamond-forming medium.

[1]Institut für Geowissenschaften, Goethe University Frankfurt, Frankfurt am Main, Germany. [2]Frankfurt Isotope and Element Research Center (FIERCE), Goethe University Frankfurt, Frankfurt am Main, Germany. [3]Department of Geoscience, University of Padova, Padova, Italy. ✉e-mail: rakipov@fierce.uni-frankfurt.de

**Table 1 | Spectroscopic results for fluid and mineral inclusions in diamond**

| Provenance | Dia. type | Mineral inclusion | Paragenetic type | LA-ICPMS | Raman shift f1 [cm⁻¹] | Intensity 1 | Raman shift f2 [cm⁻¹] | Intensity 2 | Intensity ratio (f1/f2) | Fluid distribution |
|---|---|---|---|---|---|---|---|---|---|---|
| Aikhal | | | | | | | | | | |
| A3-4 | 1aAB | omph. cpx | eclogitic | x | 664 | 490 | 808 | 335 | 1.5 | patchy |
| A3-1 | 1aAB | mgchr | peridotitc | - | 673 | 14637 | 803 | 9647 | 1.5 | enveloping |
| Internatsionalnaya | | | | | | | | | | |
| Int 3-4_1 | 1aAB | mgchr | peridotitc | x | 677 | 15235 | 808 | 12029 | 1.3 | enveloping |
| Int 3-4_2 | 1aAB | mgchr | peridotitc | - | 668 | 3921 | 799 | 2560 | 1.5 | unknown |
| Udachnaya | | | | | | | | | | |
| UD3-1_1 | 1aAB | olivine | peridotitc | x | 671 | 876 | 803 | 566 | 1.5 | patchy |
| UD3-1_2 | 1aAB | olivine | peridotitc | x | 664 | 1877 | 760 | 1160 | 1.6 | patchy |
| UD3-1_3 | 1aAB | olivine | peridotitc | n.m. | 660 | 1379 | 799 | 1246 | 1.1 | enveloping |
| UD3-1_4 | 1aAB | olivine | peridotitc | x | 668 | 9760 | 795 | 7761 | 1.3 | enveloping |
| UD3-1_5 | 1aAB | olivine | peridotitc | n.m. | n.m. | n.m. | n.m. | n.m. | n.m. | enveloping |
| UD3-2_1 | 1aAB | olivine | peridotitc | x | 660 | 1930 | 790 | 1211 | 1.6 | enveloping |
| UD3-2_2 | 1aAB | olivine | peridotitc | x | 646 | 1884 | 777 | 1098 | 1.7 | unknown |
| Siberia (u.p.) | | | | | | | | | | |
| 7994 | 1aB | olivine | peridotitc | x | 660 | 3111 | 782 | 2169 | 1.4 | enveloping |
| SOB9 | n.m. | mgchr | peridotitc | n.m. | 668 | 39822 | 799 | 27649 | 1.4 | patchy |

u.p. - unknown pipe; x - data collected; Dia. - Diamond; f1 - fluid peak 1; f2 - fluid peak 2; n.m. - not measured; omph. cpx - omphacitic clinopyroxene; mgchr - magnesiochromite
Compilation of FTIR, Raman single spot, mapping, and depth profiling results, together with the provenance for each investigated fluid and mineral inclusion from one diamond sample. The paragenetic type was assessed from the typical assemblage for each mineral inclusion[6,73]. Raman intensities of the fluid film signal (f1 lies between 646 cm⁻¹ and 677 cm⁻¹; f2 lies between 760 cm⁻¹ and 808 cm⁻¹) are background corrected for all the inclusions[22], except for A3-4, SOB9, and Int3-4_1.

The presence of the silicic fluid rim also has implications for post-entrapment modification of the associated mineral inclusion, potentially facilitating diffusive loss of elements such as Pb[16] and Rb-Sr[17]. How SFRs in gem diamonds compare to HDFs in *fibrous* diamonds and other fluids/melts from the lithospheric mantle remains unknown due to methodological challenges and the inability to chemically analyse a thin fluid layer. Conventional off-line laser ablation (LA), commonly used for HDF analysis, cannot distinguish signals from fluids closely associated with mineral inclusions, and exposing inclusions risks losing the surrounding fluid.

In this study, we overcome these challenges by performing in situ LA-ICPMS depth profiling of unexposed mineral-fluid inclusion assemblages in gem-quality diamonds from the Siberian craton. This approach, combined with micro-Raman mapping of the fluid distribution around mineral inclusions, provides the first trace-element data for SFRs, allowing direct comparison with signatures of previously studied HDFs, group 1 and 2 kimberlites, and associated PIC (Phlogopite-Ilmenite-Clinopyroxene)- and MARID (Mica-Amphibole-Rutile-Ilmenite-Diopside) metasomatic assemblages[18,19].

## Results and discussion
### Silicic fluid rims around mineral inclusions in gem diamonds
We investigated seven gem-quality diamonds from Siberia. Deconvolution of FTIR spectra using the DiaMap_v18[20,21] spreadsheet revealed five *Type IaAB* diamonds (A3-4, A3-1, Int3-4, UD3-1, UD3-2), and one *Type IaB* diamond (7994; Table 1), one diamond (SOB9) was not classified.

We identified twelve mineral inclusions larger than 80 μm with micro-Raman spectroscopy: eight olivine, three magnesiochromite, and one omphacitic clinopyroxene, all verified against the RamanCrystalHunter database[22] (Fig. 1a). All investigated mineral inclusions have a Raman-active silicic fluid rim, characterised by two broad peaks in the 646–677 cm⁻¹ and 760–819 cm⁻¹ ranges (see Fig. 1b). Similar Raman-active fluid rims were previously attributed to non-crystalline material, given that the peaks are at least five times broader than those of the mineral phases[12]. These broad peaks were previously assigned to $Si_2O(OH)_6$ dimers and $Si(OH)_4$ monomers in an aqueous fluid[14,15]. We find similar relative intensities between the

~660 cm⁻¹ and the ~800 cm⁻¹ peaks, which are consistent across all rims, with an average ratio of 1.5 (SD = 0.1), matching values previously reported (avg. 1.5, SD = 0.2; Table 1)[12]. Additionally, spectral features, including a weak broad band near 1650 cm⁻¹, previously assigned to the $H_2O$ bending mode and signal attributed to OH were observed (Supplementary Fig. 5)[12,23]. However, as both features were not observed simultaneously in one single spectrum, which is required to provide definitive evidence for a hydrous component, we interpret the fluid rim as silicic (SFR). Limitations of the spectral evidence for the hydrous nature of the SFR are further discussed in the methods.

Raman mapping and depth profiling for these Raman-active fluid rims reveal two main types of fluid distribution around mineral inclusions (Fig. 1c, d; Supplementary Figs. 2–5). In the *enveloping* type, the fluid forms an almost continuous film around the entirety of the inclusion, as observed for olivine inclusions shown in Fig. 1c, d in the blue-outlined depth profiles. In the *patchy* type, fluid occurs in isolated areas and is not detected over much of the inclusion rim in Raman maps. An example of *patchy* fluid rims around olivine inclusions is shown in Fig. 1c in the grey-outlined depth profiles. While *patchy* fluid rims may represent a limited amount of fluid, they may also result from the ~1 μm spatial resolution of micro-Raman spectroscopy in areas where the fluid rim is thinner than this.

The presence of silicic fluid rims, regardless of inclusion type (paragenesis) or diamond locality, indicates this fluid plays a fundamental role in diamond formation[12]. This is further supported by its occurrence in *Type IaAB*, *Type IaB*, and in both lithospheric[12] and sub-lithospheric[24] diamonds, which form in distinct mantle environments (depths, pressures, and temperatures). Fluid rims of similar composition surrounding silicate and non-silicate inclusions (magnesiochromite) rule out the formation of the SFR due to post-entrapment reactions or decompression exsolution, confirming that the fluid represents a residual product of metasomatic diamond-forming reactions[12].

The variable thickness of the fluid rim may represent different entrapment conditions within single diamonds and between growth pulses. Previously reported fluid rim thicknesses reach up to 3 μm[12], whereas they range from submicron (micro-Raman undetectable) up to ~5 μm here

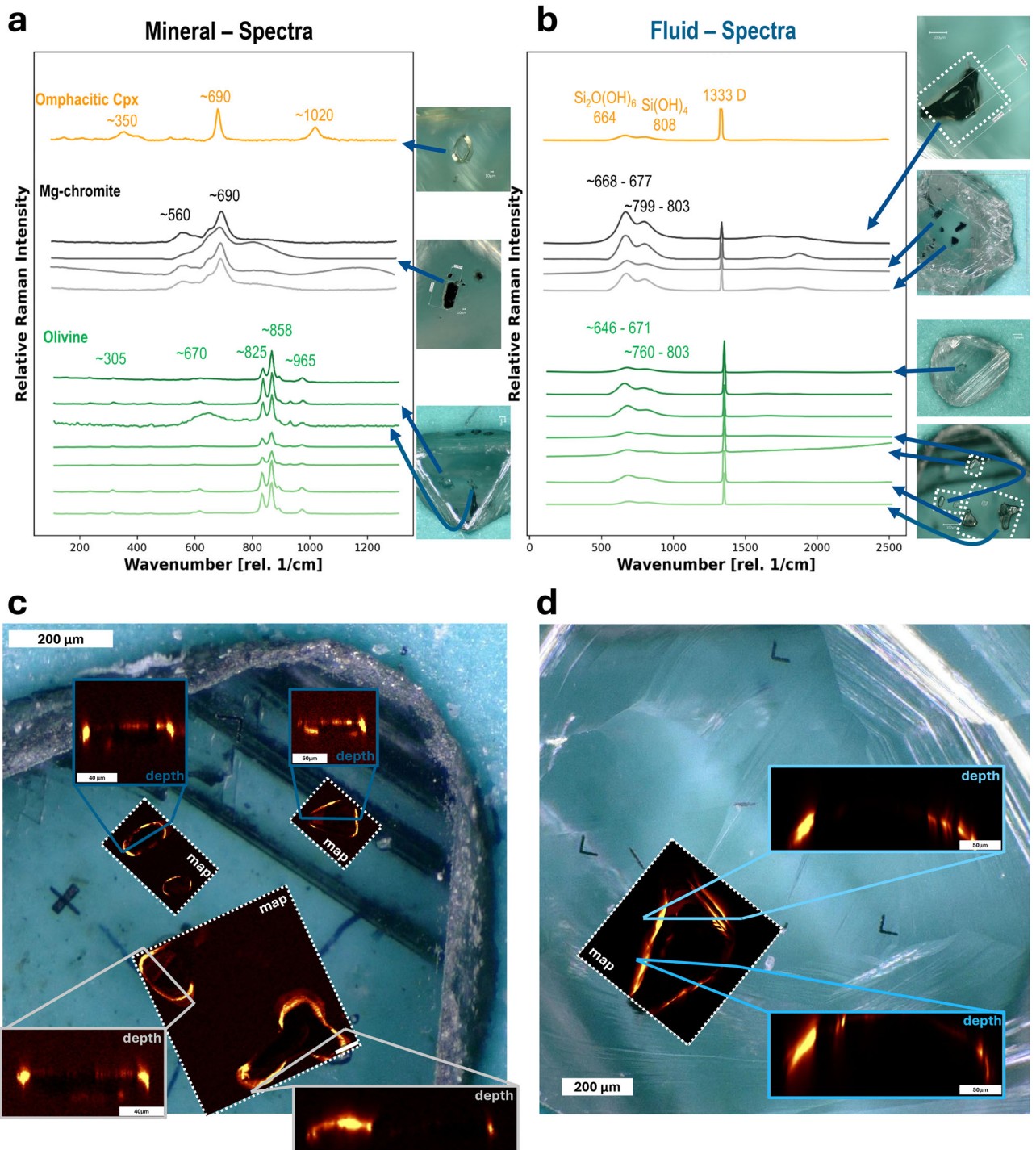

**Fig. 1 | 2D micro-Raman maps/depth-profile sections of silicic fluid rims (SFR) around mineral inclusions in diamonds.** Raman single spot spectra of (**a**) investigated mineral inclusions and (**b**) from the fluid rim around the same mineral inclusion. Arrows in (**a**, **b**) indicate the inclusion images of each spectrum. The dotted areas mark the regions mapped by Raman in (**c**, **d**). The Raman maps/depth-profiles in (**c**, **d**) were filtered for a wavenumber range, which is consistent with the SFR (Position: 660 rel. 1/cm, with a width of 100 rel. 1/cm); details regarding the map filtering are provided in methods/supplementary data. **c** Displays diamond sample UD3-1 with five investigated olivine inclusions, along with corresponding micro-

Raman maps and depth profiles. **d** Shows a map of magnesiochromite inclusion SOB9 and two measured depth profiles. In both (c, d), yellow indicates the SFR, illustrating two distribution types: an enveloping-type (grey-outlined depth profiles), where the fluid forms an almost continuous layer above the inclusion (left: UD3-1_4; right: UD3-1_3) or is detectable directly above the mineral inclusion (SOB9), and a patchy-type (grey-outlined depth profiles), where fluid occurs exclusively at the rim of inclusions and not above (UD3-1_1 and UD3-1_2). Collection parameters of Raman spectra and all individual maps/depth-profiles are provided in Supplementary Data 1a and Supplementary Figs. 2–5.

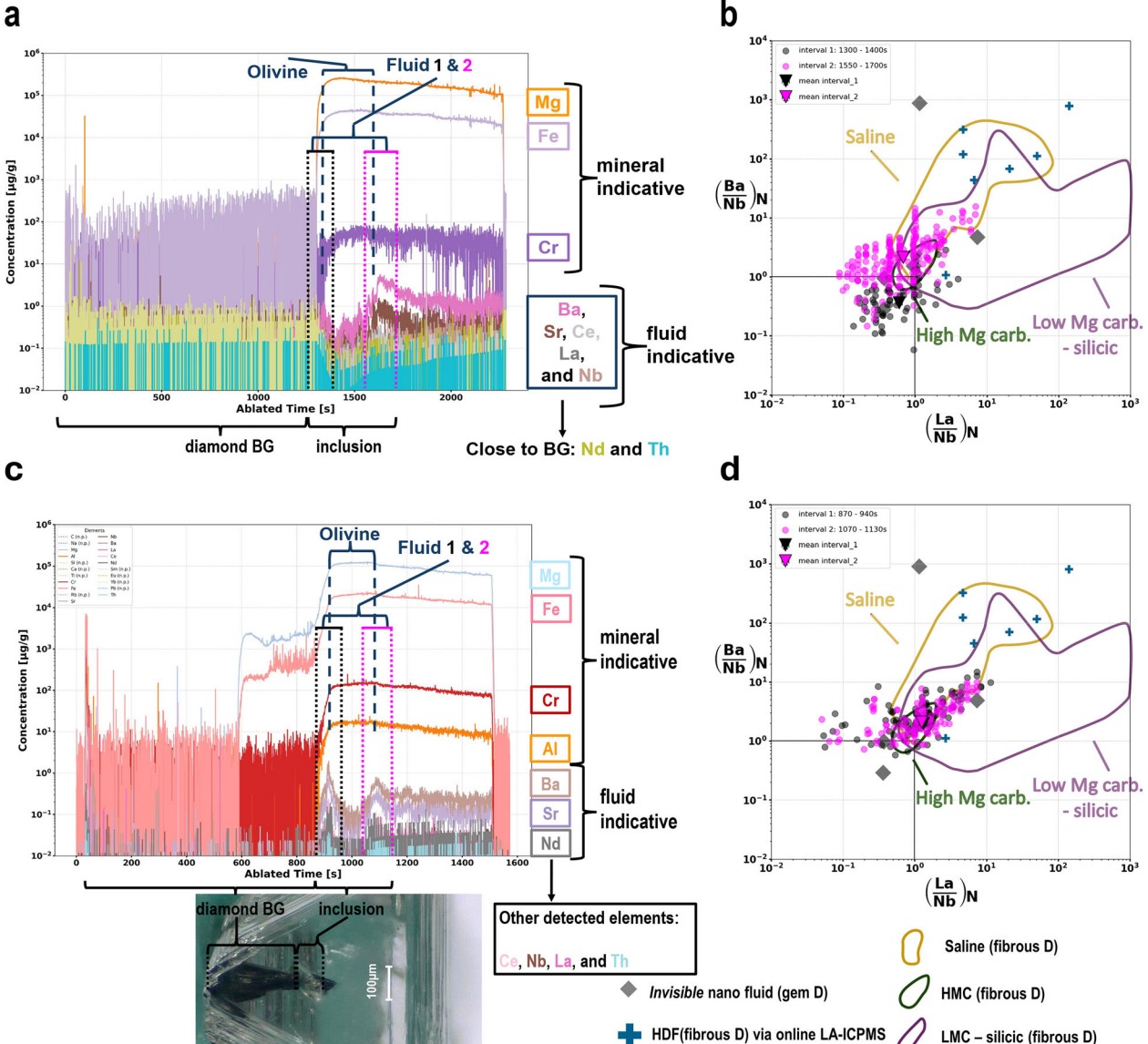

**Fig. 2 | Chemical depth profiles and PM-normalised systematics of SFR signals around olivine inclusions in diamond. a** Depth profile of sample UD3-1_4 showing the onset of signal due to the olivine-inclusion and SFR assemblage after ~1300 s of ablation. **c** Depth profile of sample 7994 reaching the inclusion after ~850 s. Below the depth profile, a post-ablation cross-cut image of sample 7994 illustrates which parts of the signal correspond to diamond background (BG) versus the inclusion assemblage. For both depth profiles in (**a, c**), blue dashed lines indicate the plateau-shaped olivine signal, while the black and magenta dotted lines mark the fluid signal above and below the mineral inclusion. Olivine is indicated by Mg, Fe, Cr, and Al; peak-shaped signals of Ba, Sr, Ce, Nb, La, Nd, and Th correspond to the fluid. **b, d** Primitive mantle-normalised[74] data extracted from the SFR-related signal of sample 7994 (**b**) and UD3-1_4 (**d**), shown in $(La/Nb)_N$_$(Ba/Nb)_N$ space. Black and magenta dots represent data from the first (above-inclusion) and second (below-inclusion) fluid intervals, triangles indicate the means. The standard error (SE = $\sigma$/sqrt(n)) is represented by error bars; if not visible, the error is smaller than the symbol size. The data points are extracted from the black and magenta dotted lines marked region in the depth profiles. Coloured fields represent different HDF compositions, typically analysed using the offline LA-method[7,9,38,47,49]. Grey diamonds show invisible-fluid compositions in gem-quality diamonds[5]. Blue crosses in (**c, d**) represent HDFs in fibrous diamond analysed with a similar LA-ICPMS approach[8]. Black lines indicate primitive mantle ratios.

(Fig. 1c, d). This variability in fluid abundance may reflect differences in diamond growth rate, as experiments suggest that inclusion entrapment generally requires rapid growth[25].

## Trace element systematics of silicic fluid rims in gem diamonds compared to HDFs and kimberlites

In situ LA-ICPMS analyses were performed by laser-ablating through the diamond host to the inclusion interface, which was previously identified as the location of the silicic fluid rim via micro-Raman spectroscopy. During the *slow* ablation procedure, a repetition rate around 2 Hz was applied, and indicative elements were monitored to distinguish signals

from the diamond host, the mineral inclusion, and the surrounding fluid rim (details in "Methods"). This approach differs from previous attempts of analysing fluids in diamond, as the resulting chemical depth profile allows in-situ temporal separation of the inclusion and fluid phase when they are chemically different in terms of their trace element compositions.

All profiles display a common pattern: an initial stable background from the diamond host followed by a plateau- or peak-shaped onset corresponding to the inclusion (Fig.2, Supplementary Figs. 6–8). The laser-ablation (LA) time of the inclusion onset (875–2800 s) indicates the distance from the diamond surface to the inclusion and serves as a proxy for position

within the diamond. The results of depth profiles can be divided into two groups based on the elements that were observed:

*Group 1: mineral-indicative*, where only major elements (Mg, Al, Ti, Cr, Fe; see methods) are detected. This pattern, observed for Int3-4_1, UD3-1_1, UD3-1_2, and UD3-2_1, indicates exclusively olivine or magnesio-chromite inclusions. Since only typical major elements of the mineral inclusion are detected, these analyses do not allow chemical distinction of the SFR and mineral phase (Supplementary Fig. 8c–f). Here, fluid rims are likely too thin to be chemically differentiated from the mineral inclusion.

*Group 2: fluid-indicative*, where both major and trace elements were detected. The depth profiles of 7994, UD3-2_2, UD3-1_4, and A3-4 reveal elements including Sr, Nb, Ba, La, Ce, Nd, and Th (See Fig. 2, and Supplementary Figs. 6, 7, 8a–c). These trace elements are typically not abundant in investigated mineral inclusions, particularly olivine[26,27], but are considered fluid-mobile[28–30], allowing mineral inclusions and SFRs to be discriminated.

For two olivine-bearing inclusions (7994, UD3-1_4), depth profiles reveal plateau-shaped signals of the mineral inclusion (Mg, Fe, Al, Cr), with peak-shaped signals due to Sr, Ba, LREEs, and Th directly above and below the mineral (Fig. 2a, c). These features are indicative of a surrounding fluid phase, consistent with the micro-Raman maps (Supplementary Figs. 3b and 4d). The Al-in-olivine thermometer[31] yields a minimum temperature of 973 °C (SD = 8 °C; calculated for 5 GPa) for sample 7994, which is consistent with a Siberian craton geotherm[32] and demonstrates our data-reduction procedure is robust (details in methods). In contrast, the fluid distribution in A3-4 and UD3-2_2 is *patchy* (Supplementary Figs. 2a and 4b), and in A3-4, a signal from the SFR cannot be distinguished from the high trace element abundances of the clinopyroxene background[6,33–35].

The signal due to an inclusion generally appears after >30 min of ablation (depths >100 μm, Supplementary Data 1a) for *mineral-indicative* depth profiles, and sooner for *fluid-indicative* profiles (<21 min; depths <100 μm). This suggests that the signal from the fluid rim is easier to detect around shallower inclusions than from deeper inclusions. The signal from deeper inclusions may be drowned out due to downhole fractionation and pit tailing or diamond graphitisation during ablation (Supplementary Fig. 10b, c, e, f; for pit tailing)[36,37].

To evaluate the chemical composition of the SFRs, data were internally standardised to mean global Si contents for the individual mineral inclusions (see methods). The fluid-related intervals in the depth profiles of 7994 and UD3-1_4 were extracted and plotted in $(La/Nb)_N$-$(Ba/Nb)_N$, $(Nd/Nb)_N$-$(Th/Nb)_N$ and $(La/Nd)_N$-$(Ce/Nd)_N$ space (Fig. 2b,d; Supplementary Fig. 9). Fluid-related trace element ratios, from peaks, above and below the olivine inclusions (reflecting SFRs), cluster together (Fig. 2b, d; Supplementary Fig. 9), indicating a fluid phase with a consistent composition. This behaviour deviates only for $(La/Nb)_N$-$(Ba/Nb)_N$ systematics of UD3-1_4. The difference between the first and second fluid intervals is attributable to the low signal intensity of the first signal, which is close to the diamond background (Figs. 2a, 3a; Supplementary Fig. 7). In terms of the $(Nd/Nb)_N$-$(Th/Nb)_N$ systematics, the first signal does not exceed the detection limit, and therefore no mean value is shown in the diagram (Fig. 3b). In spite of this, the SFR ratios generally fall close to primitive mantle (PM) values and align with high-density fluids previously reported in *fibrous* diamonds (Figs. 2, 3).

Compared to HDFs, mean SFR ratios from the fluid-related interval in sample 7994 fall within the more PM-like region of the HDF field, commonly associated with the high-Mg carbonatitic fluids[38], particularly in $(La/Nb)_N$-$(Ba/Nb)_N$ space. Similarly, SFR compositions around inclusion UD3-1_4 also plot within the high-Mg carbonatitic HDF field. In $(Nd/Nb)_N$-$(Th/Nb)_N$ space, SFR compositions lie between the saline and low Mg carbonatitic-silicic fields, and are close to PM, with a notably higher $(Th/Nb)_N$ (Supplementary Fig. 9). In terms of LREEs, SFR compositions again fall within a range near PM-like HDFs for $(La/Nd)_N$-$(Ce/Nd)_N$ (Fig. 3c, f).

Although a broad compositional overlap exists between SFRs and HDFs, direct comparison with specific HDF types requires evaluation of

major elements, and as no Mg enrichment is observed and no $CO_2$-related Raman signal is detected in the fluid rim, such a comparison cannot be resolved by the present dataset. The silicic Raman nature of the SFRs, as indicated by Raman spectroscopy, may suggest a connection to silicic HDFs, which is further supported by the enrichment in Ba and Sr, a feature also reported for saline to carbonatitic HDF[39].

The compositions of SFRs in samples 7994 and UD3-1_4 may also be compared with other metasomatic agents, such as groups 1[40,41] and 2[40,42] kimberlite melts, PIC[19,43,44] (phlogopite-ilmenite-clinopyroxene) and MARID[44] (mica-amphibole-rutile-ilmenite-diopside) parageneses, as well as calculated fluids responsible for LREE enrichment in peridotitic garnet inclusions found in diamond[45]. We observe systematic trends in $(La/Nb)_N$-$(Ba/Nb)_N$ and $(Nd/Nb)_N$-$(Th/Nb)_N$ diagrams that consistently lie in fields defined by group 2 kimberlites and MARID xenoliths and diverge from those associated with group 1 kimberlite and PIC compositions (Fig. 3). Note that group 2 kimberlite melts (orangeites) are argued to originate by melting of MARID-metasomatised mantle[18,46].

However, LREE systematics illustrated in Fig. 3c, f reveal clear deviations from most of these different metasomatic agents. Instead, the ratios of the SFR fall close to the PM-like range of HDFs (comparably flat LREE slope) and lie just below the broad field of LREE compositions observed for peridotitic garnet inclusions in diamond[45].

## Implications for the origin of silicic fluid rims

The ubiquitous presence of fluid rims around mineral inclusions suggests they play a fundamental role in diamond genesis, either as a primary growth medium or as a residual metasomatic product. Fluid rims were observed around all investigated inclusions, independent of paragenesis, mineral type, diamond type, or sample locality (Fig. 1 and Table 1). Previously, fluids were mostly reported in *fibrous* diamonds[7,9,38,47,48], along twin planes in gem-quality diamonds[10,13], or within large areas of gem-diamonds[5] with origins attributed to mantle melts, metasomatic reactions, or subduction-related fluids. The optical invisibility of the silicic fluid rim around inclusions and related analytical difficulties have limited previous investigations.

Laser ablation depth profiles around olivine inclusions 7994 and UD3-1_4 reveal a fluid-related signal distinct from the mineral-related plateau, consistent with 3D information from micro-Raman maps and depth profiles (Supplementary Figs. 3b and 4d). Trace elements (Sr, Nb, Ba, Ce, La, Nd, Th) occur exclusively in the fluid, low Nd contents, and no Th was detected around the deeper inclusion UD3-1_4, likely due to signal attenuation. All of these detected elements are not typically abundant in olivine[26,27], to the point that they are not recommended for analysis when studying olivine inclusions in diamonds[26]. Thus, we consider a fluid-derived origin for these elements. The SFR is detected above and below inclusions and is preserved during and after initial exposure with laser ablation, contrary to expectations for a residual pressurised fluid. Its persistence for >100 s of ablation further suggests an amorphous state at room temperature, which is similar to that of HDFs considered to be viscous at mantle conditions[1].

The presence of Ba and Sr anomalies align with saline HDFs attributed to a subduction-related origin[39,49]. The $(La/Nb)_N$-$(Ba/Nb)_N$ and $(Nd/Nb)_N$-$(Th/Nb)_N$ systematics match PM-like HDF compositions from *fibrous* diamonds, suggesting a similar parental source for the fluids observed in gem-quality and *fibrous* diamonds. Major-element contents of HDF-bearing diamonds reveal a continuum from LMC to silicic, and from saline to HMC compositions[48,50–52]. Trace element ratios of SFRs lie within the PM-like region, often associated with HMC compositions[38], which likely formed through interaction of saline HDFs with peridotite[49]. This is consistent with the observed Ba and Sr enrichment in the SFRs around olivine, a typical peridotitic-type inclusion. Nevertheless, the major elemental chemistry of the SFR is required to provide a more direct comparison with specific HDF compositions.

Moreover, recent observation of silicic fluids along twin planes in gem-quality diamonds[13] suggests that *invisible* nano fluid inclusions[5] likely share the same origin as the SFRs investigated in this study. This interpretation is

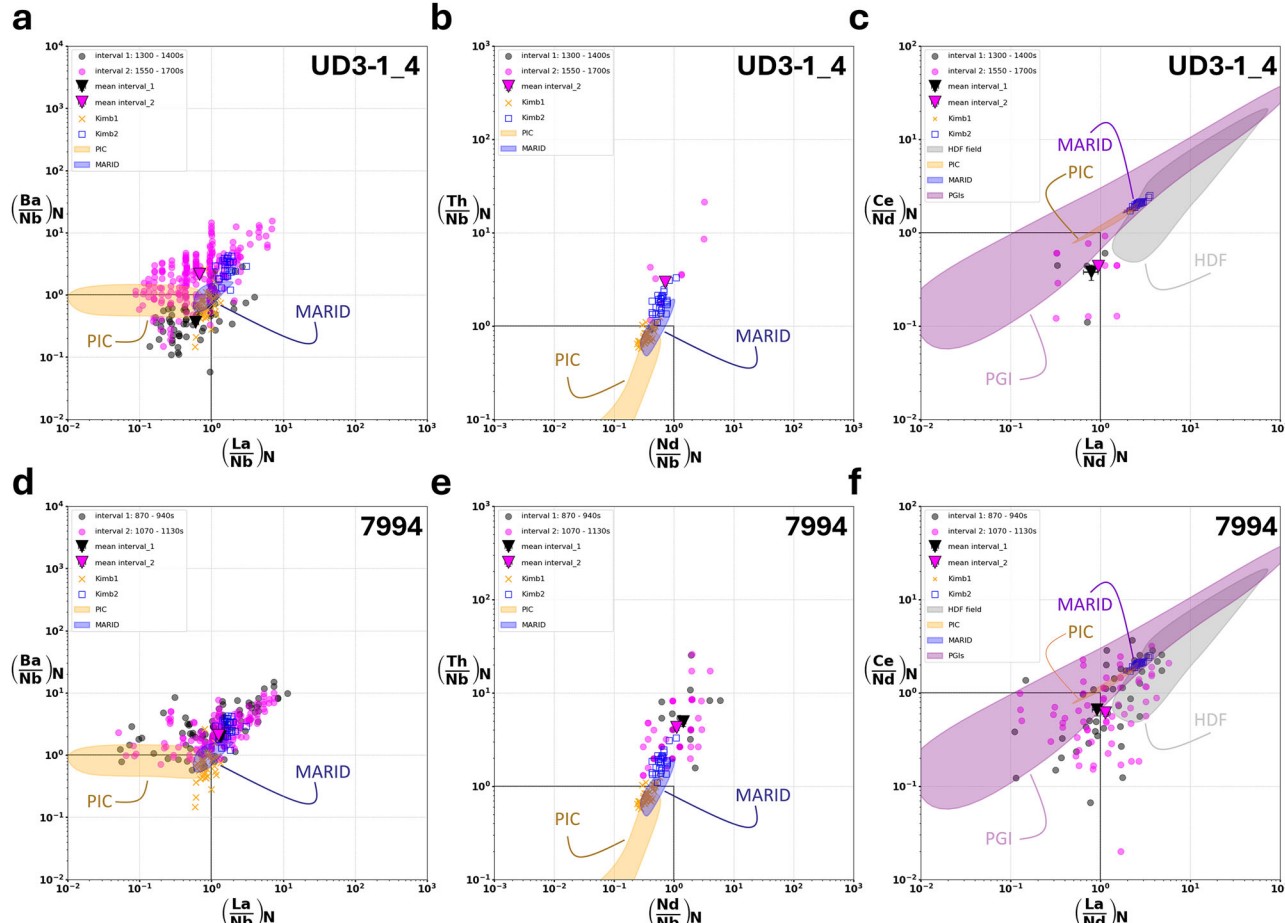

**Fig. 3 | PM-normalized trace element systematics of SFR relative to group 1 and group 2 kimberlites and PIC/MARID xenoliths.** Primitive mantle-normalised[74] data extracted from the SFR-related signal of sample UD3-1_4 (**a–c**) and 7994 (**d–f**), shown in the $(La/Nb)_N$_$(Ba/Nb)_N$, $(Nd/Nb)_N$_$(Th/Nb)_N$ and $(La/Nd)_N$_$(Ce/Nd)_N$ space. Black and *magenta* dots represent data from the first (above-inclusion) and second (below-inclusion) fluid interval, respectively, related to black and *magenta* dotted lines *marked* intervals in Fig. 2a, c. Triangles indicate their means with standard error (SE=σ/sqrt(n)) represented by error bars; if not visible, the error is smaller than the symbol size. PIC[19,43,44] and MARID[44] xenolith compositions are represented by yellow and blue fields, respectively WR-data. Yellow crosses relate to group 1 kimberlites[40,41], while open blue squares indicate group 2 kimberlites[40,42]. Black lines indicate primitive mantle ratios (PM). The purple field in **c**, **f** represents compositions of peridotitic garnet inclusions(PGI)[45] in diamond; the grey field denotes HDF compositions reported for fibrous diamonds[7,9,38].

reinforced by our observation that the SFRs fall within the broad compositional range of HDFs from *fibrous* diamonds and align with element ratios from these *invisible* nano fluids (Fig. 2 and Supplementary Fig. 9).

On the other hand, the discrepancy in physical properties, such as density (HDFs: high-density; SFR: low-density), and the fact that HDFs are found as crystallised inclusions[53–55], while the SFRs show no signs of crystalline component, suggests distinct histories for both fluid types. Some HDFs were recently interpreted to originate from low-melt/rock ratio melts of carbonated peridotite[39], while SFRs are currently considered to represent remnants of an actual low-density fluid associated with gem-quality diamond formation[12]. Fluids comprising SFRs may have formed via segregation from HDF-like melts towards a more fluid-like regime, or may reflect formation at even lower melt/rock ratios, approaching a low-density fluid phase.

Here, we also observe overall similarities in trace element systematics between SFR compositions and group 2 kimberlites and MARID xenoliths; the latter considered to represent the metasomatised mantle source of group 2 kimberlites[18]. This indicates that the SFRs may be genetically related to similar metasomatic melts. Although kimberlite melts are considered diamond-destructive due to their oxidising nature, they may promote diamond crystallisation when migrating through and reacting with more reducing domains of the lithospheric mantle[56–58]. Thus, SFRs may form under conditions of low melt/rock ratios of MARID-metasomatised mantle

domains or by segregation from a group 2 proto-kimberlitic melt. This fluid may then percolate through the peridotitic wallrock, overprinting (enriching) garnets in LREEs[34], and resulting in the PM-like LREE signature before crystalling (Fig. 3c, f) diamond through reduction.

The first chemical analyses of SFRs in this study point to a genetic relation with group 2 kimberlitic melts. Notably, comparable processes have been proposed for some HDFs, which have been interpreted as low-degree melts from carbonated peridotite[39]. In this context, the observed overlap in trace element systematics between both fluid/melt types may indicate that HDFs and SFRs originate from similar sources. However, the exact processes by which SFRs form and the extent of their genetic relation require further evaluation based on the major element composition of the SFRs.

The ubiquity of SFRs around inclusions in diamond, combined with our compositional constraints, highlights the importance of considering these structures as a post-entrapment diffusion interface, enabling element fractionation between the residual fluid and a given mineral inclusion. Fluid mobile and radiogenic elements like Th (U-Pb dating) or Sr (Rb-Sr dating)[16,59], which we detect in the fluid rim, may directly affect age determinations[17] and need to be accounted for in future investigations of mineral inclusions in diamond. The SFRs represent residual diamond-forming fluids trapped around mineral inclusions. Future studies, combining isotopic analyses of the mineral inclusions with detailed geochemical investigations of the associated fluid rims, may provide a more

comprehensive picture of the mantle domains involved in diamond formation and the associated processes by which SFRs form.

## Methods

A total of thirteen mineral inclusions were investigated from eight gem-quality diamonds. All the diamond samples originate from kimberlites that intruded the Siberian craton, including two Aikhal diamonds (A3-4 and A3-1), one Internatsionalnaya diamond (Int3-4), two Udachnaya diamonds (UD3-1 and UD3-2), and two diamonds from unknown Siberian pipes (SOB9 and 7994). Accordingly, the studied diamonds originate from at least four different kimberlitic pipes. Most of the kimberlites that host diamonds on the Siberian craton are of Palaeozoic age, emplaced between 344 and 380 Ma[60].

The diamonds range in size from 1 to 3 mm and contain single to multiple (up to eight) inclusions. The inclusions vary in diameter from the smallest at 87 μm (UD3-1_4) to the largest at 437 μm (SOB9). Inclusions smaller than 80 μm were excluded, as chemical depth profiling and distinguishing between fluid and mineral inclusions using LA-ICPMS becomes increasingly challenging at smaller scales. All investigated inclusions display rounded morphologies. Except for one diamond (Inclusions: UD3-1_3 and UD3-1_4), none were polished. All inclusions remained fully enclosed within the diamond host, allowing for in situ characterisation. It was verified via optical microscopy that any cracks in the diamond are sealed and do not extend to the surface.

The inclusion depth from the diamond surface was measured using the optical Keyence Microscope (VHX600) at the Goethe University Frankfurt. Note that these depth estimates do not account for the optical effects of diamond and its high refractive index[61]. Apparent depths range from 72 μm (7994) to 215 μm (UD3-2_1). A compilation of the inclusion depths is provided in Supplementary Data 1a; corresponding images can be found in Supplementary Fig. 1.

### Fourier-transform infrared (FTIR) and confocal micro-Raman analysis

Fourier-transform infrared (FTIR) spectra were recorded from the diamonds at the University of Padova using a Thermo Fisher Nicolet iN10 Infrared Microscope equipped with a KBr beam splitter and an LN-cooled MCT detector. Spectra were collected at a resolution of 2 cm$^{-1}$ over the range of 750–4000 cm$^{-1}$, averaging 64 scans with a total scan time of 22.4 s. Before FTIR analysis, the diamond was cleaned and mounted on a BaF$_2$ background window. To determine the extent of spatial heterogeneity in N-aggregation state and total N content due to zoning, several point spectra were collected from different regions of each diamond. The raw spectra were baseline-corrected, normalised to a thickness of 1 cm, and the N-region was deconvoluted using the Excel spreadsheet DiaMap_v18, applying standard procedures from Howell et al.[20,21]. Tabulated FTIR results are presented in Supplementary Data 1a.

Raman spectra were collected using a WITec alpha 300R confocal micro-Raman Microscope at Goethe University Frankfurt (GUF), following prior training in inclusion analysis in diamond at the University of Padova. The measurements covered a spectral range from 0 to 3900 cm$^{-1}$. A 50× objective was used, along with a 532 nm excitation laser operating at approximately 40 mW before the objective. A holographic grating with 600 grooves/mm was applied. The instrument was calibrated using an Ar-Hg spectral lamp, and its performance was verified prior to measurements based on the 1300 cm$^{-1}$ line of silicon.

For single-spot analyses, the exposure time varied between 0.1 and 0.2 s, with accumulations ranging from 5 to 20 scans. Raman spectra of mineral inclusions were compared to reference data available in the RamanCrystalHunter database[22]. Representative raw Raman spectra of the investigated mineral inclusions and silicic fluid rims (SFRs) are provided in Supplementary Data 1b. Mapping and depth profiling were performed on areas ranging from 80 × 80 μm$^2$ to 400 × 400 μm$^2$, with depth profiles extending between 50 and 400 μm. Step sizes ranged from 0.5 to 2 μm, and the exposure time for each measurement was set to 0.05 s (Supplementary Data 1a). Note that, for example, a step size of 1 μm corresponds to one point analysis per μm along a line, with lines spaced 1 μm apart. The resulting maps are then filtered for a specific wavenumber position for all collected spectra within the mapped area.

The identification of the SFR, in the Raman maps and depth profiles, was produced based on filtering intensities at characteristic wavenumber centres: 840 cm$^{-1}$ (width: 100 cm$^{-1}$), 740 cm$^{-1}$ (width: 200 cm$^{-1}$), or 660 cm$^{-1}$ (width: 100 cm$^{-1}$), depending on the coexisting inclusion type. However, certain mineral inclusions exhibit major peaks in similar wavenumber ranges as SFRs, which can complicate signal separation. For instance, omphacitic clinopyroxene (around 690 cm$^{-1}$), olivine (around 825 cm$^{-1}$ and 858 cm$^{-1}$), and magnesiochromite (a broad peak at around 690 cm$^{-1}$ with a shoulder near 800 cm$^{-1}$) overlap with key fluid rim signals. In cases where filtering a mixed signal was unavoidable, mineral-related contributions to the Raman maps could not be completely excluded, and maps and depth profiles were produced using the signal intensity at a position of 840 cm$^{-1}$ (width: 100 cm$^{-1}$). However, the resulting Raman maps still provide a clear visual distinction between fluid and mineral inclusions, supported by independent identification of the fluid rim by point analysis following the mapping (Fig. 1; Supplementary Figs. 2–5).

### Spectral features of the silicic fluid rim (SFR)

The spectral features of the SFR observed in our study are consistent with those described by Nimis et al.[12]. However, a recent experimental study demonstrated that an amorphous silicic component does not coexist with water at the pressure-temperature conditions relevant for the lithospheric mantle[62], and thus raises the question of whether the SFR can be considered hydrous. In our spectra, we occasionally observe features that have previously been attributed to a hydrous component: a weak, broad band near 1650 cm$^{-1}$ associated with H$_2$O bending and a shoulder above 3400 cm$^{-1}$ related to OH stretching. While the H$_2$O bending at 1650 cm$^{-1}$ spatially correlates with the silicic component in micro-Raman maps (Supplementary Fig. 5e, f), we could not establish the same relation for the OH stretching signal, which would be required to provide definitive evidence for a hydrous component within SFRs. Additionally, this signal (Supplementary Fig. 5a) and the analogous signal attributed to OH-stretching by Nimis et al.[12] is much broader (starting from ~2000 cm$^{-1}$) than would be expected for OH-stretching due to H$_2$O, Si$_2$O(OH)$_6$ or Si(OH)$_4$ groups. The broad shoulder above 3400 cm$^{-1}$ reported by Nimis et al.[12] and confirmed here may alternatively reflect fluorescence or an associated artefact of background correction. There appears to be no correlation between the signal attributed to H$_2$O bending and OH-stretching in the spectra. For example, where the signal attributed to H$_2$O bending is relatively intense (Supplementary Fig. 5c, d), no OH-stretching signal is observed, and vice versa (Supplementary Fig. 5a). Despite the higher sensitivity of FTIR to OH compared to Raman, attempts to detect OH-stretching or a H$_2$O bending signal in the infrared have been unsuccessful. However, it is possible this may be due to the large volume of diamond sampled by transmission FTIR analyses coupled with the small volume of the fluid rims. Thus, although our observations suggest the possibility of a hydrous contribution, they do not provide definitive evidence and consequently, more work is still required to further constrain the composition and physical properties of the fluid rims.

### Chemical LA-ICPMS depth profiles

Single spot trace elemental analysis of the fairly deep-seated (70–200 μm) mineral and fluid inclusions was performed by slow depth-profiling using laser ablation inductively coupled plasma tandem mass spectrometry (LA-ICP-MS/MS). Specifically, we utilised a Triple Quadrupole ICP-MS/MS (Agilent 8900) coupled to a custom-built Dual-Wavelength (157 and 193 nm) LA system, operated at 193 nm[63] at the Frankfurt Isotope and Element Research Centre (FIERCE) of Goethe University Frankfurt. Before measurement, the diamonds were laser-engraved with small caret symbols, cleaned with ethanol, and imaged with a Keyence Microscope. This image was then aligned with the live laser camera before the LA-ICP-MS/MS measurement, using the engraved symbols as reference points visible in the

live view of the laser ablation system. This procedure was used for orientation and improved targeting of the deep-seated mineral inclusions. This was necessary because some of the inclusions were not visible in the laser camera, and due to the large impact of the parallax effect of the off-axis viewing setup of RESOlution LA system in high refractive index materials[64].

Ablation was performed in a He atmosphere (0.3 l/min), to which Ar was added at a flow rate between 0.94 and 1.0 l/min. To enhance sensitivity and plasma stability[65,66], an additional diatomic gas, $N_2$ or $H_2$, was added for analysis of samples 7994, A3-4, Int3-4_1, UD3-1_1, UD3-1_2, UD3-2_1 (3 ml/min, $N_2$), and UD3-2_2 and UD3-1_4 (5 ml/min, $H_2$), respectively. The aerosol and carrier gas mixture flows to the ICP-MS/MS via a squid signal smoothing device[64]. Ultimately, $H_2$ is preferred due to the better Si detectability owing to the lower background at m/z = 28;29.

Laser fluence was ~12 J/cm², and the laser pulse frequency was triggered using the set to between 1.8 and 2.5 Hz using the Quadlock alignment device[67] to fire the laser per MS sweep, resulting in a repetition around 2 Hz, depending on MS setup. The LA-ICP-MS/MS system was tuned to robust plasma conditions, monitored via oxide production rates ($^{248}$ThO/$^{232}$Th below 0.5%), $^{238}$U/$^{232}$Th ~ 1, and doubly charged ion production rates (monitored via m/z = 22/44) below 0.7%.

A total of 21 elements, reduced to 16 elements by eliminating $^{85}$Rb, $^{172}$Yb, $^{208}$Pb, $^{147}$Sm, and $^{153}$Eu on day four, were selected for analysis. They were categorised into three groups: (1) elements to distinguish between the diamond host, fluid, and mineral inclusions ($^{13}$C, $^{25}$Mg, $^{27}$Al, $^{29}$Si, $^{52}$Cr); (2) elements for comparison with previously characterised fluids in *fibrous* and gem-quality diamonds ($^{23}$Na, $^{25}$Mg, $^{27}$Al, $^{29}$Si, $^{43}$Ca, $^{57}$Fe, $^{93}$Nb, $^{138}$Ba, $^{139}$La, $^{146}$Nd, $^{232}$Th); and (3) rare earth elements (REEs) and alkalis of additional interest ($^{48}$Ti, $^{85}$Rb, $^{140}$Ce, $^{147}$Sm, $^{153}$Eu, $^{172}$Yb). Following the switch to $H_2$ as additional diatomic gas, $^{28}$Si was retained and $^{29}$Si added to the element list.

Measurements were performed in single quad mode; one total sweep took between 0.4018 s and 0.5380 s, depending on the dwell times used and the number of measured masses (see Supplementary Data 2a).

For data quantification, we established the following processing procedure, following the protocol of Pettke[37], who established a routine for analysing LA-ICPMS depth-profiling data from fluid inclusions in quartz:
(a) For each analysis, intervals were defined for the blank and diamond background, as well as the mineral inclusion, and, if applicable, the surrounding fluid rim.
(b) A two-step correction procedure was applied to all calculated concentrations. First, a blank correction was performed by subtracting the mean signal of the blank (0–30 s). Second, a diamond-background correction was applied, using a measurement-dependent interval, starting from 50 s (to avoid surface contamination) and lasting up to e.g. 500 s for 7994 and 1300 s for UD3-1_4.
(c) Element concentrations were calculated using internal standardisation combined with external calibration, following the approach of Longerich et al.[68], determining the concentration of each element *i*, using:

$$C_i = C_{IS} \times \frac{C_i^{std}}{C_{ref}^{std}} \times \frac{I_i^{sample}}{I_{ref}^{sample}} \times \frac{I_{ref}^{std}}{I_i^{std}}$$

where $C_{IS}$ is the concentration of the internal standard, $C_i^{std}$ and $C_{ref}^{std}$ are the concentrations of the respective mineral inclusion and reference element in the external standard, and *I* represents measured intensities in the sample and the standard. Silicon was used as the reference element. Note that here, we use literature-derived global mean $SiO_2$ contents of corresponding mineral inclusions in diamond: olivine 41.02 wt%; omphacitic clinopyroxene 54.79 wt%; magnesiochromite 0.22 wt%. Major-element compositions of these inclusions were taken from established reference datasets[45]. Mineral inclusions were independently verified by Raman spectroscopy. NIST610 glass served as the external calibration, with NIST612 and ST.HS-G serving as secondary standards. The accuracy of secondary-standard depth profiles (up to 1303.34 s), based on comparison between measured and published

average element concentrations for NIST612 and ST.HS-G glasses are around ±10% (Supplementary Data 2a). Deviations from this are likely related to heterogeneities in NIST glasses, as observed for Mg, or due to the volatility of elements such as Pb, which makes them more sensitive to downhole fractionation[69].
(d) The concentrations for the trace element ratio plots were filtered using the Poisson-based Limit of Detection (LoD), which was calculated following Pettke et al.[70]:

$$LoD_i^{Poisson} = \frac{3.29\sqrt{R_{bkg,i} \times DT_i \times N_{an} \times (1 + N_{an}/N_{bkg})} + 2.71}{N_{an} \times DT_i \times S_i}$$

where $R_{bkg,i}$ is the mean background count rate determined from the blank interval, $DT_i$ is the dwell time of an individual element, $N_{bkg}$ and $N_{an}$ are the number of measurements in the blank and analyte intervals, and $S_i$ is the sensitivity defined as $S_i = I_i^{std}/C_i^{std}$. The use of a Poisson LoD is justified by the low background and consequently low countrates of the analysed trace elements, for which counting statistics follow a Poisson distribution rather than a normal distribution[70].

Besides the secondary standard accuracy check, this quantification procedure was also tested with the Al-in-olivine thermometer[30] for the depth profile of sample 7994, which yields a minimum temperature of 973 °C (calculated for 5 GPa; SD = 8 °C; n = 261 over 950–1060 s of the depth profile; avg. Al ~16.63 µg/g). This temperature is rather low for a lithospheric diamond, plotting in the lower end of the range in equilibration temperatures reported for peridotitic inclusions from lithospheric diamonds[71], and this condition lies just outside the experimental calibration range (1000–1300 °C) of the thermometer[31]. Further, it is not possible to know whether the olivine inclusion was originally in equilibrium in a garnet peridotite assemblage. However, this estimate represents a reasonable temperature for the Siberian craton geotherm[32] and demonstrates that our data reduction technique is robust.

It is noteworthy that the elemental composition of the fluid rims has previously not been characterised by any method and, therefore, cannot be directly internally standardised. Consequently, the quantification applied here permits only the use of trace element ratios for comparison with other studies. Supplementary Data 2a provides background-corrected, quantified data together with Poisson- and Gaussian-based LoDs for all depth profiles in Fig. 2 and Supplementary Figs. 6–8, along with details of the LA-ICP-MS/MS setup and the accuracy of secondary-standard depth profiling for each measurement day. Raw counts for all depth profiles are provided in Supplementary Data 2b.

## Data availability
Acquisition parameters for micro Raman spectroscopy and LA-ICPMS depth profiling, together with tabulated FTIR results, raw Raman spectral data for Fig. 1, and both raw count data and processed LA-ICPMS depth profile data for all profiles, are available as Supplementary Data Excel Files 1a, b and 2a, b at https://doi.org/10.5281/zenodo.19820790[72]. All Supplementary Figs. 1–10 are provided in the Supplementary Information.

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

## Acknowledgements

We are thankful for the technical support of Alexander Schmidt and Richard Albert with microscope imaging and Catharina Heckel for helpful discussions and assistance with evaluating literature data. We appreciate the insightful reviews provided by Thomas Stachel, Oded Navon, and Rondi Davies, which significantly improved the manuscript. FIERCE is financially supported by the Wilhelm and Else Heraeus Foundation and by the Deutsche Forschungsgemeinschaft (DFG: INST 161/921-1 FUGG, INST 161/923-1 FUGG and INST 161/1073-1 FUGG), which is gratefully acknowledged. This is FIERCE contribution No. 244.

## Author contributions

A.R.: creating Raman Map measurement routine, LA-ICPMS, data interpretation, writing the manuscript, conceptualisation. A.B.W.: Raman measurement, data interpretation, reviewing the manuscript, conceptualisation, funding acquisition. F.N.: Raman measurement, sample contribution, reviewing the manuscript. M.G.: Raman and FTIR analysis. M.C.D.: FTIR analysis and diamond polishing, reviewing the manuscript. M.G.P.: FTIR analysis, reviewing the manuscript. D.N.: FTIR analysis, reviewing the manuscript. T.E.: setting up LA-ICPMS depth profiling methodology, reviewing the manuscript. W.M.: conceptualisation LA-ICPMS depth profiling methodology, reviewing the manuscript, funding acquisition

## Funding

We express our gratitude to the Deutsche Forschungsgemeinschaft grant (DFG: WO 652/38-1), acquired by A.B.W., for providing financial support for the LA-ICPMS and micro-Raman analyses. This project was financially supported by European Research Council (ERC) grant Discovering **M**ineral n**A**noinclusions in **DiAM**ond (MADAM: 101199315) acquired by F.N. and ERC grant INHERIT (INHERIT: 101041620) acquired by M.G.P., which is gratefully acknowledged. Open Access funding enabled and organized by Projekt DEAL.

## Competing interests

The authors declare no competing interests.
