## [Transparent Peer Review file · Communications Earth & Environment]

Trace element systematics constrain the origin of fluids that form gem-quality diamonds

Corresponding Author: Mr Aleksandr Rakipov

This manuscript has been previously reviewed at another journal. This document only contains information relating to versions considered at Communications Earth & Environment.

Version 0:

Decision Letter:

Dear Mr Rakipov,

Your manuscript titled "Fluids in gem-quality and fibrous diamonds are compositionally similar" has now been seen by our reviewers, whose comments appear below. In light of their advice we are delighted to say that we are happy, in principle, to publish a suitably revised version in Communications Earth & Environment, provided you improve the data presentation and ensure the conclusion aligns with the evidence presented supporting a shared mantle-fluid origin for gem and fibrous diamonds.

We therefore invite you to revise your paper one last time to address the remaining concerns of our reviewers. At the same time we ask that you edit your manuscript to comply with our format requirements and to maximise the accessibility and therefore the impact of your work.

EDITORIAL REQUESTS:

****Please take care to match our formatting and policy requirements. We will check revised manuscript and return manuscripts that do not comply. Such requests will lead to delays. ****

SUBMISSION INFORMATION:

OPEN ACCESS:

Communications Earth & Environment is a fully open access journal. Articles are made freely accessible on publication. For further information about article processing charges, open access funding, and advice and support from Nature Portfolio, please visit <https://www.nature.com/commsenv/open-access>

Link Redacted

Best regards,

Céline Martin, PhD
Editorial Board Member
Communications Earth & Environment

Alireza Bahadori, PhD
Senior Editor
Communications Earth & Environment
Consulting Editor
Communications Sustainability

REVIEWERS' COMMENTS:

Reviewer #1 (Remarks to the Author):

This study is important because it provides the first in situ trace-element constraints on the thin “silicic fluid rims” (SFRs) that occur as films around mineral inclusions in gem-quality diamonds. Using micro-Raman mapping to locate the rims and slow LA-ICPMS depth-profiling through the diamond to the inclusion interface, the authors are able to separate diamond/fluid/inclusion signals and directly detect fluid-associated Sr, Nb, Ba, La, Ce, Nd, and Th in the rim material. The manuscript is also very well cited overall, and it places the new dataset clearly within prior work on diamond fluids and fibrous-diamond HDF compositional fields.

The authors primitive-mantle-normalized ratio systematics align with published high-density fluid (HDF) fields from fibrous diamonds and are interpreted to indicate a high-Mg carbonatitic affinity, supporting the broader conclusion that gem-quality and fibrous diamonds can share a common parental mantle fluid origin.

I support the overall value and publishability of the work (with minor revisions), while focusing on fixes that improve clarity, documentation, figure interpretability, and rationale and citations for key interpretive choices (e.g the (La/Nb)_N-(Ba/Nb)_N and (Nd/Nb)_N-(Th/Nb)_N plot). In addition what “major-element contents” refers to in this context and more complete reporting (how representative the two highlighted inclusions are of the full set of 13).

Below are detailed comments related to the text

Fig. 2b and 2d. Would you be able to change the color of the triangles? They are currently very difficult to see in the plots. Also, the green “High Mg carb.” line is very hard to distinguish; I recommend using a brighter, higher-contrast color. More generally, it may be worth avoiding red/green combinations because some readers will have red–green color blindness.

Line 122. I'm not sure I'm understanding what you mean by “could also reflect the ~1 μm spatial resolution of micro-Raman spectroscopy.” Could you please clarify the intended meaning and consider rewriting this sentence to make the connection explicit?

Line 135 / Fig. 1. It took me some time to interpret Figure 1. One point that added confusion for me is that the depth profiles for 1C and 1D are shown in different colors. Would you consider adjusting the color scheme (or adding a clearer key/labeling) so the figure is easier to follow at first glance?

Line 183. Could you provide a brief rationale and/or a supporting citation for why you chose to plot the data in (La/Nb)_N-(Ba/Nb)_N and (Nd/Nb)_N-(Th/Nb)_N space?

Line 186. Please add a citation at the end of this sentence to support the statement being made.

Line 189-190. Can you clarify what you mean by this? “Despite this variation, and the trend of data from low-Mg carbonatitic(LMC)-silicic compositional field overlapping the saline field,”

Line 214. Table 1 indicates that the study includes 13 inclusions, but here the discussion focuses in detail on two inclusions.

Could you add some information about what you observed for the other 11 inclusions, even briefly, so the reader has a clearer sense of how representative the two detailed examples are?

Line 224. Could you clarify what you mean by “major-element contents” here, and specify which elements you are referring to? In this study, fluids are classified using in (La/Nb)_N_(Ba/Nb)_N and (Nd/Nb)_N_(Th/Nb)_N space. Are La, Ba, Nd, Th, and Nb being treated as the “major elements” of diamond-bearing fluids, or are these ratios being used as proxies for the major-element chemistry of fibrous-diamond fluids? Explain how the two relate.

Line 224. Please provide citations supporting the statement about how these fluids match those reported for fibrous diamond.

Line 229. Could you comment on the eclogitic clinopyroxene (cpx) inclusion in your study here, and how it fits into the interpretations being discussed?

Line 235. Please add a citation supporting these two statements: “HDFs are found as partly recrystallised inclusions, while the SFR shows no signs of recrystallisation.”

Lines 189–190. Could you clarify what you mean by the following phrase: “Despite this variation, and the trend of data from low-Mg carbonatitic (LMC)–silicic compositional field overlapping the saline field,”? I’m having trouble understanding the intended logic/relationship between the variation, the trend, and the field overlap.

Extended Figure 6. Can you clarify whether the many vertical lines in the single-element plots are marking the position of the fluid film between the diamond and the inclusion? If so, could you also add these vertical lines to the larger multi-element diagram that maps all elements, so the reference position is consistent across panels?

Reviewer #2 (Remarks to the Author):

See attached file.

I apologize if the review is not coherent. It was prepared between sirens as a war is going on outside. It was not easy to concentrate on a scientific task.

Reviewer #3 (Remarks to the Author):

When Nimis et al. (2016) published their Raman-based very exciting finding of hydrous silicic fluid films around silicate and oxide inclusions in diamond, they threw a challenge at the geochemical community: figure out a way how to analyze in situ the elemental composition of fluid films with a thickness of at best a few micrometers. It took a decade but now the Frankfurt-Padua team took on the challenge and certainly succeeded. In the present manuscript they report measurements of Sr, Nb, Ba, La, Ce, Nd and Th in the fluid rims present in two of the studied diamonds (a lot more work was done, but only two samples allowed clear separation of fluid film vs inclusion contribution during depth profiling). I congratulate the team to this fantastic achievement. And given the importance of this data set for characterizing the agent of formation of monocrystalline flat faced diamonds (“gem diamonds”) and the associated metasomatic modification of deep seated cratonic peridotites, I 100% support publication in a high-impact journal, like *Communications Earth & Environment*.

Despite this unqualified support for publication, there are of course (as in any manuscript) some points that need the attention of the authors before final acceptance. I added a number of comments and requests for edits directly into the ms (attached). In addition to that, the paper needs a little more work in two aspects:

(1) The authors push an interpretation for their data that does not exactly come naturally. One almost gets the impression that already before the first analytical run, the decision was made that these fluid films have to link to the Mg-carbonatite HDFs found by the Hebrew University group (Weiss, Navon and colleagues) in fibrous diamonds. I presume, that is also the principal reason why the data are only presented in plots normalized to Nb (something that Weiss does a lot, and it makes sense for the interpretation of his data, as a negative Nb anomaly is a key ingredient of his “ribbed patterns”). From the Raman spectroscopy conducted, the fluid films do not contain carbonate (CO₂ cannot be confirmed/excluded with Raman in diamond, see Nimis et al. 2016), so why do we even discuss a high-Mg carbonatite HDF? Be bold and state that you have something new and different: a proper fluid (not HDF=melt) with (possibly hydrous) silicic character and high contents of incompatible trace elements. Then characterize the fluid, based on all the seven elements you quantified for them, not just based on ratios to Nb. For example, what is the slope in the LREE[N]? You can use the primitive mantle-normalized ratios of La/Nd and Ce/Nd to investigate that. Do they correspond to the extremely steep negative slopes predicted from trace elements in harzburgitic garnet inclusions or are they indeed more like the flatter slopes seen in HDFs in fibrous diamonds? After all this hard analytical work, your final conclusion in the abstract could be a bit bolder than “gem and fibrous diamonds share a common mantle fluid origin” (a conclusion that does not even properly fit your data).

(2) A few more requests that aim at making the entire analytical process more transparent and accessible for the reader.

- I “kind of” understand how concentrations were derived from raw counts, but not really in detail (is the Si signal measured

with the trace elements of the SFR entirely attributed to the associated silicate inclusion?). This needs to be better explained.

- The spreadsheet in the supplement gives a lot of info, but I could not find the original signals (count rates) anywhere. Only concentrations. To understand the significance and precision of, e.g., the reported Nb concentrations that are used to normalize all plotted data, original counts would be critical.
- The authors were very rigorous and investigated downhole fractionation through ablation on standards. But a critical piece of information on that is missing in the manuscript: were the pits on the standards of comparable depth to the pits on the diamonds (>100 micron)?
- Given that we deal with low counts, especially for backgrounds, the authors should recalculate their LODs using the method of Pettke (based on Poisson distributions) instead of Longerich et al. (who assumes Gaussian distributions). At least they should check and state, if that makes a difference.
- I cannot see the LOD-filtered means in Fig. 2b (if they are there they are completely masked by the data, please make visible), but in Fig. 2d these means clearly plot below the data they are calculated from. Odd. Is this really correct?
- Do the black and red data points shown in Fig. 2b and d really include (meaningless) data below LOD? Such data should not be plotted, especially as ratios.

In conclusion, I recommend acceptance of the current manuscript after fairly minor revisions.

Edmonton, March 6, 2026
Thomas Stachel

** Visit Nature Portfolio's author and referees' website at <http://www.nature.com/authors> for information about policies, services and author benefits**

Reviewer #1 (Remarks to the Author):

This study is important because it provides the first in situ trace-element constraints on the thin “silicic fluid rims” (SFRs) that occur as films around mineral inclusions in gem-quality diamonds. Using micro-Raman mapping to locate the rims and slow LA-ICPMS depth-profiling through the diamond to the inclusion interface, the authors are able to separate diamond/fluid/inclusion signals and directly detect fluid-associated Sr, Nb, Ba, La, Ce, Nd, and Th in the rim material.

The manuscript is also very well cited overall, and it places the new dataset clearly within prior work on diamond fluids and fibrous-diamond HDF compositional fields.

The authors primitive-mantle-normalized ratio systematics align with published high-density fluid (HDF) fields from fibrous diamonds and are interpreted to indicate a high-Mg carbonatitic affinity, supporting the broader conclusion that gem-quality and fibrous diamonds can share a common parental mantle fluid origin.

1: I support the overall value and publishability of the work (with minor revisions), while focusing on fixes that improve clarity, documentation, figure interpretability, and rationale and citations for key interpretive choices (e.g the $(La/Nb)_N$ $(Ba/Nb)_N$ and $(Nd/Nb)_N$ $(Th/Nb)_N$ plot). In addition what “major-element contents” refers to in this context and more complete reporting (how representative the two highlighted inclusions are of the full set of 13).

→ Regarding the representativeness of the two successful chemical analyses in the LA-ICPMS depth profiles:

Overall, we observe Raman features of this SFR around all investigated mineral inclusions, suggesting that this fluid is a ubiquitous feature of diamonds that contain mineral inclusions. The fact that the SFR was only detected around inclusions 7994 and UD3-1_4 via LA-ICPMS is most likely related to the depth of the mineral inclusions within the diamond. LA-ICPMS depth profiling is prone to signal drowning, particularly for trace elements (which are critical to identify the SFR), making the inclusion depth a key factor for successful chemical depth profiling. Another possible explanation is that the fluid varies in thickness around the mineral inclusions, as indicated by the 2D Raman maps and depth profiles. This variability may lead to lower fluid abundances in certain areas, where the laser hits the mineral-fluid interface, where the SFR remains undetected, although being present around the inclusion. This is discussed in lines 185-189. However, the two successful analyses (note that the fluid was detected both below and above the mineral inclusion) show consistent results. Based on this, we suggest that these SFRs are derived from group 2 kimberlite-like melts, potentially originating from MARID-metasomatised mantle domains. However,

we certainly cannot determine whether SFRs associated with, e.g., eclogitic mineral inclusions, would show similar or distinct trace element ratio patterns, which could reflect a more slab-derived source, compared to the peridotitic inclusions analysed here. This question requires further investigation in future studies.

Below are detailed comments related to the text

2: Fig. 2b and 2d. Would you be able to change the color of the triangles? They are currently very difficult to see in the plots. Also, the green “High Mg carb.” line is very hard to distinguish; I recommend using a brighter, higher-contrast color. More generally, it may be worth avoiding red/green combinations because some readers will have red–green color blindness.

→ Done.

3: Line 122. I’m not sure I’m understanding what you mean by “could also reflect the ~1 μm spatial resolution of micro-Raman spectroscopy.” Could you please clarify the intended meaning and consider rewriting this sentence to make the connection explicit?

→ The absence of the fluid surrounding the whole inclusion may be related to the spatial resolution of the Raman maps. The SFR, if present as a submicron-thick layer, will not be detected due to the resolution of the Raman, and not due to its absence. We have adjusted for clarification (see lines 127-128).

4: Line 135 / Fig. 1. It took me some time to interpret Figure 1. One point that added confusion for me is that the depth profiles for 1C and 1D are shown in different colors. Would you consider adjusting the color scheme (or adding a clearer key/labeling) so the figure is easier to follow at first glance?

→ Done.

5: Line 183. Could you provide a brief rationale and/or a supporting citation for why you chose to plot the data in $(\text{La}/\text{Nb})\text{N}/(\text{Ba}/\text{Nb})\text{N}$ and $(\text{Nd}/\text{Nb})\text{N}/(\text{Th}/\text{Nb})\text{N}$ space?

→ Krebs et al. (2019) used similar systematics to compare “unprecedented” fluids obtained by large area ablation of diamond with the HDFs, which is why we adopted this.

→ Furthermore, these elements were originally not expected to be detectable in olivine/Mg-chromite inclusions, but rather in a potential fluid rim, as they are generally considered fluid mobile (line 260 onwards). Therefore, they allow a distinction between the SFR and such mineral inclusions in the depth profile.

6: Line 186. Please add a citation at the end of this sentence to support the statement being made.

→ Done.

7: Line 189-190. Can you clarify what you mean by this? “Despite this variation, and the trend of data from low-Mg carbonatitic(LMC)-silicic compositional field overlapping the saline field,”

→ The systematics used here provide a first order comparison between trace element analyses of HDFs with the SFRs analysed in this study. We intended to state with Lines 189-190 that different HDF types (saline, HMC, LMC, and silicic) broadly overlap in all the trace element ratio systematics. However, as these plots were used by Krebs et al. (2019) to distinguish between their fluids and different HDF types, we adopted this approach. The revised paragraph from line 207 onwards should now be clearer. We argue that the SFRs are comparable to the HMC HDF-type, as they systematically plot in the PM-like related region of HDFs (cf. Fig. 2; Fig.3), which is associated with the HMC. However, a more robust comparison would require major element compositions.

8: Line 214. Table 1 indicates that the study includes 13 inclusions, but here the discussion focuses in detail on two inclusions. Could you add some information about what you observed for the other 11 inclusions, even briefly, so the reader has a clearer sense of how representative the two detailed examples are?

→ We attempted to analyse 8 of the 13 investigated Inclusions using the slow LA-ICPMS depth profiling. The depth of the inclusions varied between ~70 and >200 microns. The two inclusions for which the fluid rim was successfully detected are located at depths between 70 and 100 μm within the diamond. For Inclusions deeper than 100 μm , we observe signal drowning, which hinders the detection of trace elements that are essential for distinguishing between fluid and mineral inclusions.

→ Extended Figure 8 shows six depth profiles in which the fluid could not be detected, most likely because the inclusions are too deeply located within the diamond. Notably, the fluid was identified for all investigated 8 mineral-SFR assemblages by Raman mapping prior to LA-ICPMS depth profiling (cf. Extended Fig. 1-4). The reasons why the fluid is detected in only two LA-ICPMS depth profiles are also discussed in lines 185-189 of the manuscript.

9: Line 224. Could you clarify what you mean by “major-element contents” here, and specify which elements you are referring to? In this study, fluids are classified using in $(\text{La}/\text{Nb})\text{N}/(\text{Ba}/\text{Nb})\text{N}$ and $(\text{Nd}/\text{Nb})\text{N}/(\text{Th}/\text{Nb})\text{N}$ space. Are La, Ba, Nd, Th, and Nb being treated as the “major elements” of diamond-bearing fluids, or are these ratios being used as proxies for the major-element chemistry of fibrous-diamond fluids? Explain how the two relate.

- We think that we have a bit of a misunderstanding here. Weiss and coworkers have established four different HDF types, which are distinguished based on their major element contents and via ternary diagrams (e.g. in the $\text{Na}_2\text{O}+\text{K}_2\text{O} - \text{CaO}+\text{MgO}+\text{FeO} - \text{SiO}_2+\text{Al}_2\text{O}_3$ space). Here, we compare different HDFs based on their trace element ratios (all La, Ba, Nd, Th, and Nb are considered trace elements) with SFRs, in order to set them into a compositional context and relative to the major element-based HDF classification. Similar attempts, as discussed above, were previously used by Krebs et al. (2019).
- Although we provide direct chemical measurements of the silicic fluid rims, our analytical approach does not allow us to evaluate the major element characteristics of the SFR.

10: Line 224. Please provide citations supporting the statement about how these fluids match those reported for fibrous diamond.

→ Done.

11: Line 229. Could you comment on the eclogitic clinopyroxene (cpx) inclusion in your study here, and how it fits into the interpretations being discussed?

→ We have omitted this sentence from the main manuscript, as it was a bit speculative.

12: Line 235. Please add a citation supporting these two statements: “HDFs are found as partly recrystallised inclusions, while the SFR shows no signs of recrystallisation.”

→ Done. The new version can be found in line 280 onwards.

13: Lines 189–190. Could you clarify what you mean by the following phrase: “Despite this variation, and the trend of data from low-Mg carbonatitic (LMC)–silicic compositional field overlapping the saline field,”? I’m having trouble understanding the intended logic/relationship between the variation, the trend, and the field overlap.

→ See above in response to comment 7:.

14: Extended Figure 6. Can you clarify whether the many vertical lines in the single-element plots are marking the position of the fluid film between the diamond and the inclusion? If so, could you also add these vertical lines to the larger multi-

element diagram that maps all elements, so the reference position is consistent across panels?

→ Done.

Reviewer #2 (Remarks to the Author):

Reviewer: Oded Navon

The paper by Rakipov et al. provides new information on films surrounding mineral inclusions in diamonds. Such films were reported previously by Nimis et al. (2016) who measured their unique Raman spectra for 53 inclusions. The present paper provides new analyses of 13 inclusions and adds maps and depth profiles of the peaks at ~660 and ~800 cm^{-1} . The main achievement of the present paper is the first analyses of the chemical composition of the films. This is achieved by laser drilling into the films and into the inclusion below the film. They did not get a full analysis, but they do get a few important elements that allow comparison with other known diamond forming fluids. As discussed below, I think that the data and especially its presentation must be explained in much more detail. There also some conclusions that must be reconsidered. However, I do think that if the authors can solve these issues within the tight framework of the journal, the new chemical data is important and merits publication in Communications Earth & Environment.

- 1. FTIR results** – The FTIR data is missing. No spectrum, no tabulated data. Not even the nitrogen concentrations and speciation in the diamonds? The authors must provide the spectra (a supplementary table with wavenumbers and intensities of at least one spectrum for each diamond and, hopefully, more than one), along with Figures of the spectra in an excel spreadsheet.

In addition, have the authors tried to check if they see a water signal in the olivine itself?

- We include the diamond classification only for completeness, mentioning it only briefly in the main manuscript and noting the absence of H_2O related signal in the acquired spectra. The primary focus of this study is on the newly obtained trace elemental data for the SFR, rather than on spectroscopic data, which are used to set the chemical results in the context of existing literature (e.g., this is the same fluid that was reported by Nimis et al.,2016).
- We agree that a more detailed investigation of the presence or absence of a hydrous component in the SFR would be worth looking at in future studies with greater detail. However, this lies beyond the scope of the present work.
- Tabulated FTIR results are now provided in Supplementary Information 1a.

- 2. Raman results** – First, the Raman data is not in the paper supplements. I could not find a table of the spectra. I do not expect all the spectra of the depth profiling, but I do expect to see a few representative spectra for each inclusion and its fluid film as tables of intensity vs. wavenumber. For sure, I would like to

see the “unfiltered spectra” for those that appear in Fig. 1 and in Extended Fig. 5.

→ Done.

3. In figure 1, it is not clear what is presented. Are the presented spectra of the films single-spot analyses, or an average of the whole “Map” as written on Fig. 1? What is the “Filter” that is subtracted from the “Map”? Does the filter include the olivine peaks that do not show in the fluid spectra? I find it hard to imagine that they can be avoided completely. But I saw in Nimis et al. (2016) that they get spectra with no olivine signal as well.

→ The presented spectra are all representative single-spot analyses. This has now been clarified in the figure caption.

→ Regarding the filter, we are not entirely sure what is meant by “subtracted from the map”. The Raman maps and the filtering work as follows: each map consists of individual point analyses (see settings in supplementary information 1a) acquired with a stepsize of 1-2 μm ; for a stepsize of 1 μm , one point analysis is performed per μm along a line, with lines spaced 1 μm apart. The resulting map is filtered for a specific wavenumber position for all collected spectra in the area of the map (e.g. 660 rel. $1/\text{cm}$, with a width of 100 rel. $1/\text{cm}$; for SFR peak 1). The map therefore displays the highest intensity for the selected wavenumber region (in yellow), corresponding to the location of the SFR around mineral inclusions. We have now clarified that in the methods section.

→ We observe the same silicic fluid rim surrounding the mineral inclusions as described by Nimis et al. (2016), and as it is shown in Figure 1, the SFR is detectable without spectra mixing with the surrounding mineral inclusions.

4. I failed to understand the part in “Methods” that explains the filtering. See my comment for line 54 in “methods”.

→ See above (3. Comment).

5. Are spectra collected on different spots around the inclusion similar or there are differences?

→ The spectra of the SFR remain consistent. This was verified using oscilloscope mode (live measurement) and by double-checking individual spectra from the fluid rim regions after Raman map acquisition.

6. Next, the authors dismiss the interpretation of Nimis et al. (2016) for the nature of the films (lines M63-66 in “Methods”. On one hand, I do not understand why is it important if a hydrous fluid with dissolved $\text{Si}(\text{OH})_4$ or $\text{Si}_2\text{O}(\text{OH})_6$ exist in high T? The fluid clearly have many other ions dissolved in it and changes the speciation during cooling. Another question is whether such a fluid remains unchanged when trapped at high T together with diamond, olivine, chromite, garnet, or cpx, or whether it must react with the minerals? On the other hand, I think that the fluid is far from water + silicic acid or its dimmer. It probably

contains many other ions. The peaks at 600-850 cm^{-1} are probably due to silica and water, but it can still be a saline fluid with many ions dissolved in it. It is strange that the monomer-dimer ratio is constant (with 660/800 intensity ratio of 1.5) while the fluid is in contact with different minerals. I think this is a bigger obstacle than these species stability at high T. Also, if the 1650 cm^{-1} line is due to water, then we should see a consistent intensity ratio between the 1650/660 or the 1650/3400 lines. The authors tell us that they do not see the latter, they do not provide the data or discuss the first correlation except for claiming that the two (the 1650 and 660 cm^{-1} lines) correlate spatially (Line M69).

- We are not sure whether we understand the point being made here. We do not argue that the SFR cannot contain cations. We verify solely the Raman observations of Nimis et al., which postulated that this fluid is a silicic remnant of a diamond-forming fluid. We take their argument as a starting point for our interpretation. Regarding the spatial correlation of 1650/660, this is a feature we occasionally observe, which means that it is not consistent for all SFR spectra and can be very well seen in the spectra presented in Figure 1b. In fact this is one of the reasons why we try here to report doubts about the hydrous nature of the SFR as discussed in the methods section, we do not see evident consistencies of the hydrous signal and the typical SFR related signal. However, our study rather focuses on the here presented new trace elemental data acquired for the SFR, than the presented Raman or FTIR data, which is rather used to set the chemical results into the context present in the literature. We agree that this point however, is worth looking into in future studies that may focus on the spectral features of the SFR.
 - Further, there may be different compositions of the fluid around different mineral inclusions. However, Raman spectroscopy would not be the method of choice to verify this. So far, we have only been able to detect the fluid around olivine inclusions with the LA-ICPMS slow depth profiling. Other inclusions were deeply located within the diamond (>100 micron), which led to signal drowning and prevented the detection of the thin surrounding fluid rim. We suggest that this may be resolvable with follow up studies investigating the fluids trace element composition around e.g. eclogitic mineral inclusions, which should show different systematics. However, this is not resolvable with dataset present for this study but will require future work.
7. But then, if the authors do not accept this identification, Why do they keep treating the films as silicic fluid rims (SFR)? What in the data that suggests the presence of water + silica? More important, they see no peaks of carbonates (at 1050-1100 cm^{-1}), CO_2 or CH_4 . So, we are left with no identification of the material in the fluid film. This should be stated out loud in the main paper.
- Our observations are generally consistent with those of Nimis et al., supporting the interpretation that these fluids represent silicic fluid remnants from the diamond forming reaction. However, we would like to note that the hydrous

component, which is described in that study, can be considered doubtful given the discussion in the methods section.

→ Regarding the HMC expected spectral features: We have now added a clarification that no CO₂ signal is observed in the Raman spectra, which would be expected for fluids similar to those of the HMC high-density fluids (cf. from line 207 onwards).

8. Last, I examined the position of the two Raman lines in the data of both Nimis et al. (2016) and the present set (see below). The two sets broadly agree and overlap. I notice that the positions of the two peaks correlate positively. I could not find any note of it in any of the papers. I think that it is a useful observation in trying to identify the source of these peaks. I suggest to check whether there is a correlation between the position of the two Raman lines and that of the olivine itself (which may hint whether the shift is the result of internal pressure). But, of course, it may be due to other causes. As can be seen in the figure, It is not a function of the type of included mineral. It is strange that the speciation of the material in the rim is not affected by the nature of the mineral next to it.

Have the authors tried to monitor the internal pressure within the olivine inclusions based on the shift of their Raman lines and whether it correlates with position of the 660 and 800 cm⁻¹ lines.

→ This is a good observation. However, elastic barometry is beyond the scope of this study, as we have focused on providing chemical evidence for the SFRs. Our interpretation is based on existing literature as discussed above, which to date consistently describes these fluids as silicic. We suggest further investigations on that in future more spectroscopy-based studies; however, we cannot provide a definitive answer to this point.

9. **LA-ICPMS data** – The reader needs more explanations about how the data was acquired and processed. For example, how was the LoD calculated along the run. The reference to Longerich et al. (1996) is not enough. How does one determine the standard deviation of the blank during the run? Do you use the

average value from the initial (or initial and final) blank measurement(s) for the whole analysis?

→ The data procedure routine has now been revised, and the exact formula used is specified. We generally use the initial blank for quantification. However, both the initial and final blanks are now reported in a separate supplementary Excel file containing the counts for each depth profile.

10. If so and if, in some measurements, the reading on the sample is a continuous zero, why does the LoD varies with time? I could not understand even after going to Longerich et al. for help.

→ We found an error in our previous calculations. The standard deviation (used for calculating the LoD) was calculated dynamically based on 32 data points around one individual measurement sweep in the depth profile, rather than being based on the initial blank. This led to the artificial overestimation of the LoDs, as they have reflected the variability of the data around each measurement point. When calculated based on the initial blank, both Poisson- and Gaussian-based LoD calculations are lower. The corrected values are now reported in the revised files.

11. What is the LoD in the Excel tables of the supplement? Is it the limit of detection, or the data after subtracting the limit of detection? I need more detailed explanation as footnotes to the tables. An example of the calculations of the LoD will be helpful.

→ Done. However, we are still not entirely sure what is meant by “data after subtracting the limit of detection”. In our data procedure routine, the LoD is only used as a filtering value for the plotted data and is not subtracted from the measured/calculated value.

12. Another mysterious feature is the signal of carbon, sodium, and calcium during analysis. Next I refer mostly to the example of 7994, but the situation and the problems are similar in other diamonds as well.

Carbon: How come the carbon signal goes to zero when the inclusion is penetrated? There is still a lot of diamond around and especially above. The conical shape of the excavated volume (Fig. 1 and Extended data Figure 10) suggests that more diamond is consumed. What is the reason for the rise in the carbon LOD when the inclusion is reached? The signal drops to zero, so what can be the reason for the peak in the LOD?

Sodium: In sample 7994, at t=33-42 s we see the surface sodium decays. Then when the SFR is penetrated we see no rise in sodium. On the contrary, it drops to zero at t=870 s. There must be more sodium in the SFR than in the diamond. What is the reason for the peak in LoD at 900-950 s? The signal is 0, so why does the LoD change?

Calcium. Very similar to sodium. There should be Ca in the fluid and in the olivine. So why don't we see it?

Silicon: What about Si? On one hand you claim that this is the element used for calibrating the concentration, but on the other hand it equals zero for the whole analysis of 7994. Why don't you get some readings on the olivine?

→ Regarding all four elements (C, Si, Ca, Na), we generally observe a high background in our system. The fluid concentrations can easily be drowned by this background. We suggest future analysis be done with a more sensitive ICPMS set-up for these elements in order to evaluate the major element systematics of the fluid. However, this was not part of our study, considering also the very low Dwell times used for the detection of these elements. We originally intended to use them as tracers for the fluid rim, which was unsuccessful. We have now added a "<BG" for all calculated concentrations, which are below blank or diamond background to the supplementary Excel file containing the processed data, as we think that this adds to the confusion here. The values for the above-mentioned elements appear to mostly be below the detected system or diamond background, which points to them not being meaningful for the overall interpretation of the SFRs composition.

13. Aluminum: It behaves as opposite to Ca. What may be the reason to the different behavior of calcium and aluminum? Why does the LoD has a peak at ~900-930 s? I ask because this where the Ba and the incompatible elements show their peaks, so we must understand the reason for the presence or absence of peaks.

→ We have found an error in the LoD calculations. See the response to comment 10 for clarification. The methods now include a revised and more informative description of the data procedure routine.

14. Magnesium: What is the reason for the rise after ~600 s? You cannot simply ignore it. Why only Mg, Fe and Cr jump, but not Al (which remains at 0)?

→ The rise at 600 s is potentially related to an olivine-microinclusion closely above the targeted olivine inclusion, which was too small to have a detectable amount of SFR. Also, in the case of 7994, we have ablated inside the diamond in two time periods without taking the diamond out of the laser ablation cell, as in the first, we have underestimated the time required to reach the inclusion. We kept on ablating inside the diamond further, which resulted in the analysed time period presented here. This may have caused an initial piercing of the inclusion, which is highlighted in the plateau signal at 600 s. During further ablation, the pit then started to widen, leading to the detection of the SFR.

15. Nb: All the signals are below the LoD. Why is that so? It is clear that we see clear signals at 870-940 and 1040-1110. The blank is lower than 0.0005 so why is the LoD so high.

→ We found an error in our LoD calculations. See above in the response to comment 10 for clarification.

16. La, Ce, Sr, and Th: The same question as for Nb.

→ See above.

17. In summary: The presentation of data must be improved.

a. All data should be included in the tables and more should be plotted in the supplements and Methods.

→ Done.

b. The processing of data should be explained better in a way that allow the reader (and the reviewers) to use it and to reproduce the data.

→ Done.

c. Some questions as to the ability of the author to constrain the species responsible for the Raman data must be answered and if it is not possible to identify them, it should be stated clearly.

→ We have revised this section to clarify that there are doubts regarding to the hydrous but not the silicic nature of the fluid.

d. The different behavior of various elements during the LA profile must be explained.

→ This was due to the major elements being below the diamond/system(blank) background, which has now been clarified.

e. The processing of the LA data should be explained.

→ Done.

18. The title of the paper

The similarity of the material in the rim to HDF is not clear enough to justify the present title. Basically, using Raman, the paper maps the position of the rim material around, above and below the inclusions. This is important and remarkable. Using depth profiling by LA-ICPMS the authors reveal the presence of a few incompatible elements in the rim material. The authors do not identify the nature of the fluid and its main constituent. As a matter of fact, the authors do not know whether it is a low-density or high-density fluid, or whether it is a fluid at all (probably it is). They also do not have the major element composition of the rim material, but have many incompatible trace elements, which again, is remarkable. The title has to reflect the results.

- The title has been adjusted.
- Regarding the low-density features of the SFR. As highlighted by Nimis et al., 2016 in the SRXTM images, these fluids indeed show low-density characteristics.

19. Abstract

Same as the title: The data clearly indicate the presence of material enriched in certain incompatible trace elements. From here to “suggesting a high-Mg carbonatitic composition” is a long way. Please see my specific comments on lines 48 and 50.

- We have now phrased this more carefully. Indeed, postulating an HMC composition requires more evidence from the major elements. Nevertheless, HMC have been associated with PM-like compositions, which is interesting given that this is what we observe for the SFR in terms of the trace element ratios presented in this work.

Specific comments:

(the line numbers refer to the line numbers in the submitted version. The prefix M referred to the line number in the “Methods” section.)

20.5: The address of Muller should be 1,2 not 12.

- Done.

21.48: The concentrations of the incompatible trace elements are elevated relative to primitive values, but their average ratios (e.g., the triangles in Fig. 1 b and D) are close to PM and even lower.

- Done.

22.50: I do not like the phrasing of this sentence. The $(La/Nb)_N$ $(Ba/Nb)_N$ and $(Nd/Nb)_N$ $(Th/Nb)_N$ systematics of the fluid do align with those of high-density fluids (HDFs) in fibrous diamonds. However, other types of HDFs can also produce such ratios. There are many other obstacles to identification of the material in the rims as High-Mg carbonatitic (HMC)-HDFs.

- The SFRs are consistent with the general field of HDFs, which we consider a valid observation. As mentioned above, we agree that the classification of specific HDF-types requires major element data. This has now been clarified in the revised version of the manuscript.

a. There are no peaks of carbonate in the Raman spectra.

- We have adjusted that, as we solely detect silicic signals in the Raman spectra, and this is not what would be expected for the HMC.

b. The major elements do not show any of the expected characteristics of HMC-HDF. For example, there is no Na peak when the rim is penetrated. The Ca

signal decays quickly after the penetration. Also, while Ba and Sr behave similarly, La peak rises only 20 seconds later together with Al.

- See above in responses to comments 12 and 13 for clarification regarding the major elements. Both Ca and Na exhibit a high system background in the LA-ICPMS. Combined with relatively low dwell times used for Ca and other major elements, this makes it unsuitable for constraining the major element systematics of the SFR.
- Regarding the non-simultaneous behaviour of trace elements, we disagree with this interpretation. Ba, Sr, La and other trace elements detected in the fluid show coherent behaviour in the depth profiles (cf. enlarged plot below for 7994).

23.66: How do you know that the signals observed by Krebs et al. are from “invisible nano-inclusions”? They are “invisible” and were never observed by any imaging methods. So their size is unknown.

- This is the terminology used to describe the fluid in Krebs et al. (2019) (cf. Last paragraph of their introduction). We follow the original terminology here.

24.69: You summarize three studies. Jablon and Navon observed the inclusions they analyze by EPMA imaging. The inclusions in Smith et al. are visible under the microscope (they are microns to tens of microns in size). So how come the summary is that “these fluids are optically invisible”? Perhaps they are simply rare inclusions?

→ We agree that both the inclusions analysed by Smith et al. and Jablon and Navon can be considered as rare features that can occasionally be found in gem diamonds. Regarding the SFR, this is, however, not necessarily the case, as these fluids have now been reported by several independent studies; they can now rather be argued as a common fluid feature around mineral inclusions in diamond.

25.78: The HDF are NOT “thought to have partly recrystallized after entrapment”. They were observed by TEM to be fully or almost fully recrystallized after entrapment.

→ Adjusted.

26.86 and 90: The relation between the material in the rims of the inclusions and HDFs is not fully solved in this paper as well. The authors clearly advance our understanding, but they do not fully “overcome these challenges” and the exact composition of the rims is not yet determined.

→ We think that there is a misunderstanding here. Our study shows that the challenge of chemically analysing the SFR can be overcome, which is the point addressed in this paragraph. At the same time, we do agree that there is still a lot of work left to do, as our chemical characterisation is solely based on trace elements. This limitation is now stated in the manuscript, for example, by noting that major element analyses of the SFR would be required to provide comparison to specific HDF types.

27.95: I suggest not to use the expression “Silicic fluids”.

→ This is necessary to remain with the established terminology, introduced by Nimis et al., who originally referred to them as hydrous silicic fluids.

28.96-98: This is the only reference to the FTIR results.

→ Yes, we only provide the diamond type for completeness, since FTIR seems not to be suitable for directly targeting the silicic fluid rim.

29.105: There are 13 inclusions and 4 mgchr, not 12 and 3.

→ Done.

30.110: The identification of the peaks at 660 and 800 cm⁻¹ by Nimis et al. (2016) is given here without any comment. However, in “Methods” the authors cast some doubt on this identification. I join this doubt, based on the constant position of the two peaks and their constant intensity ratio, regardless of the mineral phase with which the rim material is in contact. I think that such doubt should be expressed in the main text as well.

→ We have now added a sentence addressing the doubts.

- 31.** 126-129: While the first part of the sentence sounds fine: “Fluid rims of similar composition surrounding silicate and non-silicate inclusions (magnesiocromite), rule out post- entrapment reactions or decompression exsolution”. But then, how does that confirms “that the fluid represents a residual product of metasomatic diamond-forming reactions”? Why would reaction with eclogites be the same as with peridotite?
- This was a phrasing error on our side. We intended to state that the observation of the SFR independent of the mineral type rules out a post-diamond formation-related origin of these fluids.
- 32.** 133: Can the variability in the thickness of the rims be the result of the angular mismatch between the trapped mineral and the growing diamond?
- Possible, but this cannot be resolved by the present dataset.
- 33.** Fig. 1: a. There are no scales on the photos to the right of the RAMAN spectra in Fig 1. a and B. Alternatively, list the width of each figure. b. In panel (b) what do you mean by Map-filter? What is “Map” and what is “filter”? They are not defined anywhere? Or, I suddenly suspect you just overuse the “-“ sign and mean “Map filter” – the spectral range from which you collect the signal for the “heat maps” (another strange term). c. In the “heat maps”, why are the inclusions in panel (c) darker than the diamond? Both have no peak at 550-900 cm⁻¹?
- Scale bars are included in the small figures, these images are not really suitable for providing the scales; For this reason, all diamonds and enlarged inclusions are shown with scales in the extended figure 1.
- Regarding the Raman map filter, please see our response to comment 3.
- Regarding the different colour schemes of the maps, this is related to the large map of inclusions UD3-1_1 and UD3-1_2, being processed with a different wavenumber filter range (position 740; width 200 [rel. 1/cm]) compared to the other maps (position 660; width 100 [rel. 1/cm]). We have adjusted this by filtering all maps in Figure 1 to the same wavenumber range.
- 34.** 176-180: The whole paragraph is hard to understand. You distinguish between “Mineral indicative” and “Fluid indicative”, but then shift to deeper versus shallow. Also, here you list panels B, C, E, F in Fig. 10 as examples of pit tailing, but in the caption of Fig. 10 you list only b and C while E and F are considered successful.
- To successfully analyse the fluid inclusion using the LA-ICPMS depth profiling, it is essential that the inclusion assemblage (mineral inclusion together with surrounding SFR), is located as close to the surface as possible. Therefore, we distinguish between fluid-indicative profiles (where the fluid could be analysed) and mineral-indicative profiles (where only the mineral inclusion was detected), in order to provide an estimate of the cut-off depth and ablation time for future studies.

→ The figures are referred to as examples of LA pit tailing, resulting in the conical shape of the ablated pit from the side view. However, Fig.10 also shows successful analysis in the sense that the mineral inclusion located within the diamond was successfully hit by the LA-ICPMS. “Unsuccessful” refers to case, where the inclusion is located too deep within the diamond that it cannot be reached by LA-ICPMS. The ablation pit thins out before reaching the inclusion. We have added another sentence to the caption for clarification.

35.181: You claim to standardize to Si, but in 7994, Si=0 throughout the analysis.

→ This was unclearly presented in the previous version of the supplementary information. As Si is used as an internal standard and given the formula that is now provided in the methods section, the calculated Si concentrations will always reflect the concentration assigned to the internal standard, which was listed in the file.

→ In the new supplementary_information 2b, we additionally provide the raw counts for each depth profile. These data show that Si is detected in the sample, but not in the SFR. Given that the Si is a high background element, this does not necessarily mean that Si is absent in the fluid rim. Rather, the analytical setup is not sufficiently sensitive to detect such high background major elements, particularly considering the micron-scale thickness of the SFRs.

36.183: What is the uncertainty involved with the ratios for individual measurement? It should be indicated on the graphs in Fig. 2 panels b and d.

→ The standard error ($SE=SD/\sqrt{N}$) is now added to the figure.

37.185: They fall near the primitive mantle values. Both below and above. Delete the “above”.

→ Done.

38.187-192: In Fig. 2, panels b and d, about 80% of the small area spanned by the high-Mg carbonatitic HDFs overlaps with those of the low-Mg carbonatitic to silicic HDFs and saline HDFs. This actually prevents the identification of the rim material with one of the four end-members of the HDFs. An argument against such identification is the lack of carbonate Raman line. Examination of the classic composition of the four endmembers reveals that while the molar CO₂/H₂O ratio of the HMC-HDFs is almost 2, it is only ~0.2 in the silicic and saline HDFs, making them better candidates for the rim material. The silicic component is rich in SiO₂ and may form the Raman peaks. A saline endmember is better in coexistence with olivine or chromite. I do not see any reason to prefer the HMC-HDF.

→ This part of the discussion is now adjusted, providing a more general comparison with the HDFs and not focussing on specific HDF types anymore. However, we find that it is still worth mentioning that the SFR plot within rather

PM-like compositions when compared to the HDF, which has been correlated with HMCs in prior studies.

39. Fig. 2. Panel (a) – Which element is the light-yellow color and which is the cyan? Both La and Nb are written in black instead in the color of the data. Panel (b) – What is the composition of the HDF in the diamonds analyzed by Tomlinson et al. (cyan +). What do you mean by “LoD filtered mean”? In the excel file, each element receives two columns: Nb [$\mu\text{g/g}$] and LOD_Nb [$\mu\text{g/g}$]. do you subtract the LOD from the signal? But the LOD is commonly higher than the signal. So what exactly do you do? Panel (c) – Why the Mg and Fe jump at ~ 600 s? You must explain what you see rather than just ignore it. If the inclusion is reached at 870 s and lasts until ~ 1130 s, what happens between 1130 and 1600 s? During the first 870 s the beam penetrated ~ 250 μm . Why it only penetrated ~ 30 μm during the last 470 s? This should be discussed in “Methods”. Also, why is the signal continues long after the inclusion is passed?

→ The colour labelling has now been adjusted.

→ “LoD filtered mean” has also been adjusted. All the data shown in the figures is above the LoD.

→ See the response to comment 10 for clarification on the adjustments being made regarding the LoD calculations.

→ Regarding the ablation behaviour of diamond: We observe strong pit tailing when ablating through the diamond. This leads to an initially thin pit, which widens the more pulses of ablation are performed. This is why the olivine signal persists after we already ablated through the inclusion. The pit widens further and releases new olivine into the system.

→ Regarding the detection of Fe and Mg at 600s: see in response to comment 14.

40.211: Ref. 5 do not describe inclusions along twin planes in gem-quality diamonds. It does not belong here. It should be 10,13.

→ Done.

41.221: “The presence of Ce suggests a subducted crustal component”. Ce always appear together with other REEs. Only an anomalous concentration can attest to a subducted crustal component.

→ Agree.

42.222: I agree that high level of Ba and Sr are commonly related to saline fluids. Mershon et al. (2025, Mineral Petrol) have reported a saline diamond with extremely high S and Ba.

→ Agree.

43.225-229: See comments on lines 187-192. As Weiss and coworkers argued many times, the trace element patterns are decoupled from the major element composition, so they cannot be used to identify the HDF endmember. The major

element data obtained for the major elements is not good enough to be used for identification.

→ Agree.

44.233: But the position on a Ba/Nb vs La/Nb is not unique. I think that kimberlites also fall there.

→ We agree and have now extended our discussion to a comparison with kimberlitic melts.

45.234: How was the density of the SFRs determined?

→ We assume the density based on the SRXTM observations of Nimis et al. (2016), who have shown that these fluids display low-density features.

46.241-243: So here the SFRs need to be more viscous, like the HDFs. I agree with this interpretation of the persistence of the SFR during ablation. But then, why argue for lower density than the HDFs?

→ We consider it a distinctive feature that SFRs do not exhibit any crystallised phases, which may indicate a lower density. See also SRXTM images in Nimis et al. for direct evidence supporting the SFR being low-density.

47.244-249: Good final paragraph.

→ Thanks.

Materials, methods and data protocol

48.M28: Why not convert the apparent depth into real depth by multiplying by 2.4 (the refractive index of diamonds)? It is true that this is an approximation, but it is a good one and gives a better feeling of the dimensions.

→ We mention the systematic error in our depth estimates (added also in the supplementary information file); this can be corrected for easily. We prefer to report the measured values.

49.M32-39: FTIR – Please provide spectra, tabulated data, concentrations of nitrogen as A and B centers, intensity of platelets and the 3107 line.

→ Tabulated FTIR results are now provided. See response to comment 1.

50.50. -54: What do you mean by “filtered” How do you “filter”? I really do not understand what you did.

→ See above in response to comment 3.

51.M64-66: Why should we care about speciation of Si in hydrous fluid at elevated temperatures? If the SFR are not hydrous, how do you explain the Raman peaks at ~1650 cm⁻¹?

→ The occasional peaks at 1650 cm⁻¹ point towards the SFR being hydrous. We only state here doubts regarding the general hydrous nature of this fluid

remnant, as the spectral evidence for it being hydrous is not consistent over all collected spectra. Reasons for that need to be further investigated in future spectroscopic studies.

52.M67-85: The discussion here cast doubt on the interpretation of the Raman signals. If you are not sure whether there is a hydrous component in the SFR, then how do you get the 660 and 800 cm^{-1} signals? They are interpreted as $\text{Si}(\text{OH})_4$ and $\text{SiOSi}(\text{OH})_6$ in water. If this is not the case, then what are the two peaks? Now you are no longer sure that the SFR are silicic fluid rims. So what are they? Add to that the lack of CO_2 or carbonate related bands and you lose the connection to the HDFs. You have to end this section with your best guess about the nature of the rims, and explain what constraints you have to substantiate your guess.

→ We find it an observation that is worth stating, that the SFR is not consistently showing evidence of being hydrous. This, however, does not invalidate the ubiquity of the SFR Raman spectral evidence provided in this study. We have now tried to clarify the manuscript, of what evidently is present (the silicic component) and what is occasionally present in the Raman spectra (the hydrous feature).

→ Our best guess is that it is a silicic fluid rim.

53.M110: Check the number of element and correct them. They are different in the text and in the table.

→ Done.

54.M133: How did you use Si in the case of 7994, where $\text{Si}=0$ for the whole run?

→ See above in response to comment 35.

55.M139: I failed to find the formula in Longerich et al. (1996). The limits of detection listed in the Excel table are higher than the signal, at least for 7994, so how do you perform stage (e)?

→ Done. The method section, which describes the LA-ICPMS data procedure routine, has now been adjusted.

56. Table S1 Rows 6 and 12 do not present the “Depth of inc. in μm ” but rather, as can be understood from line 50 in Methods.

→ Adjusted and noted that the depth estimates are not corrected for the high refractive index of diamond.

57. Table S4

How was the 2SD calculated? Is it the individual SD, or the SD of the mean (divided by the square root of the number of measurements)?

- Used formula for calculation of the SD: $SD = \sqrt{\frac{\sum(x_i - m)^2}{N}}$; x_i – individual value, m – mean, N – number of measurement points.
- We are not entirely sure what is meant by “SD of the mean (divided by the square root of the number of measurements)”.

Reviewer #3 (Remarks to the Author):

When Nimis et al. (2016) published their Raman-based very exciting finding of hydrous silicic fluid films around silicate and oxide inclusions in diamond, they threw a challenge at the geochemical community: figure out a way how to analyze in situ the elemental composition of fluid films with a thickness of at best a few micrometers. It took a decade but now the Frankfurt-Padua team took on the challenge and certainly succeeded. In the present manuscript they report measurements of Sr, Nb, Ba, La, Ce, Nd and Th in the fluid rims present in two of the studied diamonds (a lot more work was done, but only two samples allowed clear separation of fluid film vs inclusion contribution during depth profiling). I congratulate the team to this fantastic achievement. And given the importance of this data set for characterizing the agent of formation of monocrystalline flat faced diamonds (“gem diamonds”) and the associated metasomatic modification of deep seated cratonic peridotites, I 100% support publication in a high-impact journal, like *Communications Earth & Environment*.

Despite this unqualified support for publication, there are of course (as in any manuscript) some points that need the attention of the authors before final acceptance. I added a number of comments and requests for edits directly into the ms (attached). In addition to that, the paper needs a little more work in two aspects: (1) The authors push an interpretation for their data that does not exactly come naturally. One almost gets the impression that already before the first analytical run, the decision was made that these fluid films have to link to the Mg-carbonatite HDFs found by the Hebrew University group (Weiss, Navon and colleagues) in fibrous diamonds. I presume, that is also the principal reason why the data are only presented in plots normalized to Nb (something that Weiss does a lot, and it makes sense for the interpretation of his data, as a negative Nb anomaly is a key ingredient of his “ribbed patterns”). From the Raman spectroscopy conducted, the fluid films do not contain carbonate (CO₂ cannot be confirmed/excluded with Raman in diamond, see Nimis et al. 2016), so why do we even discuss a high-Mg carbonatite HDF? Be bold and state that you have something new and different: a proper fluid (not HDF=melt) with (possibly hydrous) silicic character and high contents of incompatible trace elements. Then, characterize the fluid, based on all the seven elements you quantified for them, not just based on ratios to Nb. For example, what is the slope in the LREE[N]? You can use the primitive mantle-normalized ratios of La/Nd and Ce/Nd to investigate that. Do they correspond to the extremely steep negative slopes predicted from trace elements in harzburgitic garnet inclusions or are they indeed more like the flatter slopes seen in HDFs in fibrous diamonds? After all this hard analytical work, your final conclusion in the abstract could be a bit bolder than “gem and fibrous diamonds share a common mantle fluid origin” (a conclusion that does not even properly fit your data).

→ We have now plotted the data in the recommended space and included literature data for different kimberlite types, as well as MARID and PIC xenoliths. These are interpreted to reflect fluid/melt metasomatised lithospheric mantle.

We also provide LREE related trace element ratio plots. The results are discussed in the manuscript (from lines 226 and 287 onwards).

→ Regarding the comparison relative to the HDFs, the similar trace element systematics observed in comparison with SFRs still suggest at least a source-related link between the two fluid/melt types (Fig.3 c,f).

(2) A few more requests that aim at making the entire analytical process more transparent and accessible for the reader.

• I “kind of” understand how concentrations were derived from raw counts, but not really in detail (is the Si signal measured with the trace elements of the SFR entirely attributed to the associated silicate inclusion?). This needs to be better explained.

→ The description of the data procedure routine has now been revised. Si is generally a high background element in the LA-ICPMS system and is therefore not suitable to be detected in the SFR, given its micron-scale size. We attribute all the Si to the detected mineral inclusions, cf. supplementary information 2b for counts, which align with the mineral plateau.

• The spreadsheet in the supplement gives a lot of info, but I could not find the original signals (count rates) anywhere. Only concentrations. To understand the significance and precision of, e.g., the reported Nb concentrations that are used to normalize all plotted data, original counts would be critical.

→ Done.

• The authors were very rigorous and investigated downhole fractionation through ablation on standards. But a critical piece of information on that is missing in the manuscript: were the pits on the standards of comparable depth to the pits on the diamonds (>100 micron)?

→ This is indeed an important comment. We did not measure pit depths in the standards after the analyses. However, based on the ablation time (1000s-1250s; cf. supplementary information 2a), we assume that the resulting pit depths in the standards are comparable to those of the diamond analyses, while acknowledging potential differences in ablation behaviour between the two materials.

• Given that we deal with low counts, especially for backgrounds, the authors should recalculate their LODs using the method of Pettke (based on Poisson distributions) instead of Longerich et al. (who assumes Gaussian distributions). At least they should check and state, if that makes a difference.

→ We have recalculated the LoDs based on both Gaussian and Poisson distributions. Both values can be found within the LA-ICPMS supplementary material.

• I cannot see the LOD-filtered means in Fig. 2b (if they are there, they are completely masked by the data, please make visible), but in Fig. 2d these means clearly plot below the data they are calculated from. Odd. Is this really correct?

→ We have double checked our background correction and have identified an error. Previously, the correction was based on the entire Diamond-Background for each depth profile, including the surface contamination. As a result, values close to background (and this is the case for the first fluid interval in UD3-1_4; Fig.2b), showed an increased scatter, leading to the offset in the mean.

• Do the black and red data points shown in Fig. 2b and d really include (meaningless) data below LOD? Such data should not be plotted, especially as ratios.

→ During the recalculating of the LoDs, we have identified an error in the previous approach. The LoD (Gaussian) had been calculated dynamically, which led to an overestimation of the LoD, particularly in the region of the fluid rim. All data currently shown in the plots is above the LoD based on Poisson distribution, which is recommended for low background elements (Pettke et al., 2012), and is therefore considered more appropriate for filtering all trace elements that are used in our interpretation.

In conclusion, I recommend acceptance of the current manuscript after fairly minor revisions.

Edmonton, March 6, 2026

Thomas Stachel

Review of the paper: "Fluids in gem-quality and fibrous diamonds are compositionally similar" by Rakipov et al.

Reviewer: Oded Navon

The paper by Rakipov et al. provides new information on films surrounding mineral inclusions in diamonds. Such films were reported previously by Nimis et al. (2016) who measured their unique Raman spectra for 53 inclusions. The present paper provides new analyses of 13 inclusions and adds maps and depth profiles of the peaks at ~ 660 and ~ 800 cm^{-1} . The main achievement of the present paper is the first analyses of the chemical composition of the films. This is achieved by laser drilling into the films and into the inclusion below the film. They did not get a full analysis, but they do get a few important elements that allow comparison with other known diamond forming fluids.

As discussed below, I think that the data and especially its presentation must be explained in much more detail. There also some conclusions that must be reconsidered. However, I do think that if the authors can solve these issues within the tight framework of the journal, the new chemical data is important and merits publication in *Communications Earth & Environment*.

Major comments

Data presentation must be improve substantially. This is true for all the analytical data: FTIR, Raman and LA-ICPMS.

FTIR results – The FTIR data is missing. No spectrum, no tabulated data. Not even the nitrogen concentrations and speciation in the diamonds? The authors must provide the spectra (a supplementary table with wavenumbers and intensities of at least one spectrum for each diamond and, hopefully, more than one), along with Figures of the spectra in an excel spreadsheet.

In addition, have the authors tried to check if they see a water signal in the olivine itself?

Raman results –First, the Raman data is not in the paper supplements. I could not find a table of the spectra. I do not expect all the spectra of the depth profiling, but I do expect to see a few representative spectra for each inclusion and its fluid film as tables of intensity vs. wavenumber. For sure, I would like to see the "unfiltered spectra" for those that appear in Fig. 1 and in Extended Fig. 5.

In figure 1, it is not clear what is presented. Are the presented spectra of the films single-spot analyses, or an average of the whole "Map" as written on Fig. 1? What is the "Filter" that is subtracted from the "Map"? Does the filter include the olivine peaks that do not show in the fluid spectra? I find it hard to imagine that they can be avoided completely. But I saw in Nimis et al. (2016) that they get spectra with no olivine signal as well.

I failed to understand the part in "Methods" that explains the filtering. See my comment for line 54 in "methods".

Are spectra collected on different spots around the inclusion similar or there are differences?

Next, the authors dismiss the interpretation of Nimis et al. (2016) for the nature of the films (lines M63-66 in "Methods". On one hand, I do not understand why is it important if a hydrous fluid with dissolved $\text{Si}(\text{OH})_4$ or $\text{Si}_2\text{O}(\text{OH})_6$ exist in high T? The fluid clearly have many other ions dissolved in it and changes the speciation during cooling. Another question is whether such a fluid remains unchanged when trapped at high T together with diamond, olivine, chromite, garnet, or cpx, or

whether it must react with the minerals? On the other hand, I think that the fluid is far from water + silicic acid or its dimer. It probably contains many other ions. The peaks at 600-850 cm^{-1} are probably due to silica and water, but it can still be a saline fluid with many ions dissolved in it. It is strange that the monomer-dimer ratio is constant (with 660/800 intensity ratio of 1.5) while the fluid is in contact with different minerals. I think this is a bigger obstacle than these species stability at high T. Also, if the 1650 cm^{-1} line is due to water, then we should see a consistent intensity ratio between the 1650/660 or the 1650/3400 lines. The authors tell us that they do not see the latter, they do not provide the data or discuss the first correlation except for claiming that the two (the 1650 and 660 cm^{-1} lines) correlate spatially (Line M69).

But then, if the authors do not accept this identification, Why do they keep treating the films as silicic fluid rims (SFR)? What in the data that suggests the presence of water + silica? More important, they see no peaks of carbonates (at 1050-1100 cm^{-1}), CO_2 or CH_4 . So, we are left with no identification of the material in the fluid film. This should be stated out loud in the main paper.

Last, I examined the position of the two Raman lines in the data of both Nimis et al. (2016) and the present set (see below). The two sets broadly agree and overlap. I notice that the positions of the two peaks correlate positively. I could not find any note of it in any of the papers. I think that it is a useful observation in trying to identify the source of these peaks. I suggest to check whether there is a correlation between the position of the two Raman lines and that of the olivine itself (which may hint whether the shift is the result of internal pressure). But, of course, it may be due to other causes. As can be seen in the figure, it is not a function of the type of included mineral. It is strange that the speciation of the material in the rim is not affected by the nature of the mineral next to it.

Have the authors tried to monitor the internal pressure within the olivine inclusions based on the shift of their Raman lines and whether it correlates with position of the 660 and 800 cm^{-1} lines.

LA-ICPMS data – The reader needs more explanations about how the data was acquired and processed. For example, how was the LoD calculated along the run. The reference to Longerich et al. (1996) is not enough. How does one determine the standard deviation of the blank during the run? Do you use the average value from the initial (or initial and final) blank measurement(s) for the whole analysis? If so and if, in some measurements, the reading on the sample is a continuous zero, why does the LoD varies with time? I could not understand even after going to Longerich et al. for help.

What is the LoD in the Excel tables of the supplement? Is it the limit of detection, or the data after subtracting the limit of detection? I need more detailed explanation as footnotes to the tables. An example of the calculations of the LoD will be helpful.

Another mysterious feature is the signal of carbon, sodium and calcium during analysis. Next I refer mostly to the example of 7994, but the situation and the problems are similar in other diamonds as well.

Carbon: How come the carbon signal goes to zero when the inclusion is penetrated? There is still a lot of diamond around and especially above. The conical shape of the excavated volume (Fig. 1 and Extended data Figure 10) suggests that more diamond is consumed. What is the reason for the rise in the carbon LOD when the inclusion is reached? The signal drops to zero, so what can be the reason for the peak in the LOD?

Sodium: In sample 7994, at t=33-42 s we see the surface sodium decays. Then when the SFR is penetrated we see no rise in sodium. On the contrary, it drops to zero at t=870 s. There must be more sodium in the SFR than in the diamond. What is the reason for the peak in LoD at 900-950 s? The signal is 0, so why does the LoD change?

Calcium. Very similar to sodium. There should be Ca in the fluid and in the olivine. So why don't we see it?

Aluminum: It behaves as opposite to Ca. What may be the reason to the different behavior of calcium and aluminum? Why does the LoD has a peak at ~900-930 s? I ask because this where the Ba and the incompatible elements show their peaks, so we must understand the reason for the presence or absence of peaks.

Magnesium: What is the reason for the rise after ~600 s? You cannot simply ignore it. Why only Mg, Fe and Cr jump, but not Al (which remains at 0)?

Silicon: What about Si? On one hand you claim that this is the element used for calibrating the concentration, but on the other hand it equals zero for the whole analysis of 7994. Why don't you get some readings on the olivine?

Nb: All the signals are below the LoD. Why is that so? It is clear that we see clear signals at 870-940 and 1040-1110. The blank is lower than 0.0005 so why is the LoD so high.

La, Ce, Sr and Th: The same question as for Nb.

In summary: The presentation of data must be improved.

- a. All data should be included in the tables and more should be plotted in the supplements and Methods
- b. The processing of data should be explained better in a way that allow the reader (and the reviewers) to use it and to reproduce the data.
- c. Some questions as to the ability of the author to constrain the species responsible for the Raman data must be answered and if it is not possible to identify them, it should be stated clearly.
- d. The different behavior of various elements during the LA profile must be explained.
- e. The processing of the LA data should be explained

The title of the paper

The similarity of the material in the rim to HDF is not clear enough to justify the present title. Basically, using Raman, the paper maps the position of the rim material around, above and below the inclusions. This is important and remarkable. Using depth profiling by LA-ICPMS the authors reveal the presence of a few incompatible elements in the rim material. The authors do not identify the nature of the fluid and its main constituent. As a matter of fact, the authors do not know whether it is a low-density or high-density fluid, or whether it is a fluid at all (probably it is). They also do not have the major element composition of the rim material, but have many incompatible trace elements, which again, is remarkable. The title has to reflect the results.

Abstract

Same as the title: The data clearly indicate the presence of material enriched in certain incompatible trace elements. From here to “suggesting a high-Mg carbonatitic composition” is a long way. Please see my specific comments on lines 48 and 50.

Specific comments (the line numbers refer to the line numbers in the submitted version. The prefix M referred to the line number in the “Methods” section.

5: The address of Muller should be 1,2 not 12.

48: The concentrations of the incompatible trace elements are elevated relative to primitive values, but their average ratios (e.g., the triangles in Fig. 1 b and D) are close to PM and even lower.

50: I do not like the phrasing of this sentence. The $(La/Nb)_N$ $(Ba/Nb)_N$ and $(Nd/Nb)_N$ $(Th/Nb)_N$ systematics of the fluid do align with those of high-density fluids (HDFs) in *fibrous* diamonds. However, other types of HDFs can also produce such ratios. There are many other obstacles to identification of the material in the rims as High-Mg carbonatitic (HMC)-HDFs.

- a. There are no peaks of carbonate in the Raman spectra.
- b. The major elements do not show any of the expected characteristics of HMC-HDF. For example, there is no Na peak when the rim is penetrated. The Ca signal decays quickly after the penetration. Also, while Ba and Sr behave similarly, La peak rises only 20 seconds later together with Al.

66: How do you know that the signals observed by Krebs et al. are from “invisible nano-inclusions”? They are “invisible” and were never observed by any imaging methods. So their size is unknown.

69: You summarize three studies. Jablon and Navon observed the inclusions they analyze by EPMA imaging. The inclusions in Smith et al. are visible under the microscope (they are microns to tens of microns in size). So how come the summary is that “these fluids are optically invisible”? Perhaps they are simply rare inclusions?

78: The HDF are NOT “thought to have partly recrystallized after entrapment”. They were observed by TEM to be fully or almost fully recrystallized after entrapment.

86 and 90: The relation between the material in the rims of the inclusions and HDFs is not fully solved in this paper as well. The authors clearly advance our understanding, but they do not fully “overcome these challenges” and the exact composition of the rims is not yet determined.

95: I suggest not to use the expression “Silicic fluids”.

96-98: This is the only reference to the FTIR results.

105: There are 13 inclusions and 4 mgchr, not 12 and 3.

110: The identification of the peaks at 660 and 800 cm^{-1} by Nimis et al. (2016) is given here without any comment. However, in “Methods” the authors cast some doubt on this identification. I join this doubt, based on the constant position of the two peaks and their constant intensity ratio, regardless of the mineral phase with which the rim material is in contact. I think that such doubt should be expressed in the main text as well.

126-129: While the first part of the sentence sounds fine: “Fluid rims of similar composition surrounding silicate and non-silicate inclusions (magnesiocromite), rule out post- entrapment reactions or decompression exsolution”. But then, how does that confirms “that the fluid represents a residual product of metasomatic diamond-forming reactions”? Why would reaction with eclogites be the same as with peridotite?

133: Can the variability in the thickness of the rims be the result of the angular mismatch between the trapped mineral and the growing diamond?

Fig. 1: a. There are no scales on the photos to the right of the RAMAN spectra in Fig 1. a and B. Alternatively, list the width of each figure. b. In panel (b) what do you mean by Map-filter? What is “Map” and what is “filter”? They are not defined anywhere? Or, I suddenly suspect you just overuse the “-” sign and mean “Map filter” – the spectral range from which you collect the signal for the “heat maps” (another strange term). c. In the “heat maps”, why are the inclusions in panel (c) darker than the diamond? Both have no peak at 550-900 cm^{-1} ?

176-180: The whole paragraph is hard to understand. You distinguish between “Mineral indicative” and “Fluid indicative”, but then shift to deeper versus shallow. Also, here you list panels B, C, E, F in Fig. 10 as examples of pit tailing, but in the caption of Fig. 10 you list only b and C while E and F are considered successful.

181: You claim to standardize to Si, but in 7994, Si=0 throughout the analysis.

183: What is the uncertainty involved with the ratios for individual measurement? It should be indicated on the graphs in Fig. 2 panels b and d.

185: They fall near the primitive mantle values. Both below and above. Delete the “above”.

187-192: In Fig. 2, panels b and d, about 80% of the small area spanned by the high-Mg carbonatitic HDFs overlaps with those of the low-Mg carbonatitic to silicic HDFs and saline HDFs. This actually prevents the identification of the rim material with one of the four end-members of the HDFs. An argument against such identification is the lack of carbonate Raman line. Examination of the classic composition of the four endmembers reveals that while the molar $\text{CO}_2/\text{H}_2\text{O}$ ratio of the HMC-HDFs is almost 2, it is only ~ 0.2 in the silicic and saline HDFs, making them better candidates for the rim material. The silicic component is rich in SiO_2 and may form the Raman peaks. A saline endmember is better in coexistence with olivine or chromite. I do not see any reason to prefer the HMC-HDF.

Fig. 2. Panel (a) – Which element is the light-yellow color and which is the cyan? Both La and Nb are written in black instead in the color of the data. Panel (b) – What is the composition of the HDF in the diamonds analyzed by Tomlinson et al. (cyan +). What do you mean by “LoD filtered mean”? In the excel file, each element receives two columns: Nb [$\mu\text{g/g}$] and LOD_Nb [$\mu\text{g/g}$]. do you subtract the LOD from the signal? But the LOD is commonly higher than the signal. So what exactly do you do? Panel (c) – Why the Mg and Fe jump at ~ 600 s? You must explain what you see rather than just ignore it. If the inclusion is reached at 870 s and lasts until ~ 1130 s, what happens between 1130 and 1600 s? During the first 870 s the beam penetrated ~ 250 μm . Why it only penetrated ~ 30 μm during the last 470 s? This should be discussed in “Methods”. Also, why is the signal continues long after the inclusion is passed?

211: Ref. 5 do not describe inclusions along twin planes in gem-quality diamonds. It does not belong here. It should be 10,13.

221: "The presence of Ce suggests a subducted crustal component". Ce always appear together with other REEs. Only an anomalous concentration can attest to a subducted crustal component.

222: I agree that high level of Ba and Sr are commonly related to saline fluids. Mershon et al. (2025, Mineral Petrol) have reported a saline diamond with extremely high S and Ba.

225-229: See comments on lines 187-192. As Weiss and coworkers argued many times, the trace element patterns are decoupled from the major element composition, so they cannot be used to identify the HDF endmember. The major element data obtained for the major elements is not good enough to be used for identification.

233: But the position on a Ba/Nb vs La/Nb is not unique. I think that kimberlites also fall there.

234: How was the density of the SFRs determined?

241-243: So here the SFRs need to be more viscous, like the HDFs. I agree with this interpretation of the persistence of the SFR during ablation. But then, why argue for lower density than the HDFs?

244-249: Good final paragraph.

Materials, methods and data protocol

M28: Why not convert the apparent depth into real depth by multiplying by 2.4 (the refractive index of diamonds)? It is true that this is an approximation, but it is a good one and gives a better feeling of the dimensions.

M32-39: FTIR – Please provide spectra, tabulated data, concentrations of nitrogen as A and B centers, intensity of platelets and the 3107 line.

M52-54: What do you mean by "filtered" How do you "filter"? I really do not understand what you did.

M64-66: Why should we care about speciation of Si in hydrous fluid at elevated temperatures? If the SFR are not hydrous, how do you explain the Raman peaks at $\sim 1650 \text{ cm}^{-1}$?

M67-85: The discussion here cast doubt on the interpretation of the Raman signals. If you are not sure whether there is a hydrous component in the SFR, then how do you get the 660 and 800 cm^{-1} signals? They are interpreted as Si(OH)_4 and SiOSi(OH)_6 in water. If this is not the case, then what are the two peaks? Now you are no longer sure that the SFR are silicic fluid rims. So what are they? Add to that the lack of CO_2 or carbonate related bands and you lose the connection to the HDFs. You have to end this section with your best guess about the nature of the rims, and explain what constraints you have to substantiate your guess.

M110: Check the number of element and correct them. They are different in the text and in the table.

M133: How did you use Si in the case of 7994, where $\text{Si}=0$ for the whole run?

M139: I failed to find the formula in Longerich et al. (1996). The limits of detection listed in the Excel table are higher than the signal, at least for 7994, so how do you perform stage (e)?

Table S1 Rows 6 and 12 do not present the “Depth of inc. in μm ” but rather, as can be understood from line 50 in Methods.

Table S4

How was the 2SD calculated? Is it the individual SD, or the SD of the mean (divided by the square root of the number of measurements)?

Table S5

Fluids in gem-quality and fibrous diamonds are compositionally similar

Aleksandr Rakipov^{1,2,3*}, Alan B. Woodland^{1,2}, Fabrizio Nestola³, Matilde Galie^{1,3}, Martha G. Pamato³,
Davide Novella³, Maxwell C. Day³, Tobias Erhardt^{1,2}, Wolfgang Müller^{1,2}

¹*Institut für Geowissenschaften, Goethe University Frankfurt, 60438 Frankfurt am Main, Germany*

²*Frankfurt Isotope and Element Research Center (FIERCCE), Goethe University Frankfurt, 60438 Frankfurt am Main, Germany*

³*Department of Geoscience, University of Padova, 35131 Padova, Italy*

**Corresponding author*

Authors Contribution:

Aleksandr Rakipov: *Creating Raman Map measurement routine, LA-ICPMS, data interpretation, writing the manuscript, conceptualisation*

Alan B. Woodland: *Raman measurement, data interpretation, reviewing the manuscript, conceptualisation, funding acquisition*

Fabrizio Nestola: *Raman measurement, sample contribution, reviewing the manuscript*

Matilde Galie: *Raman and FTIR analysis*

Maxwell C. Day: *FTIR analysis and diamond polishing, reviewing the manuscript*

Martha G. Pamato: *FTIR analysis, reviewing the manuscript*

Davide Novella: *FTIR analysis, reviewing the manuscript*

Tobias Erhardt: *setting up LA-ICPMS depth profiling methodology, reviewing the manuscript*

Wolfgang Müller: *conceptualisation LA-ICPMS depth profiling methodology, reviewing the manuscript, funding acquisition*

**Abstract**

Diamonds crystallise from fluids/melts circulating in ~~the~~ Earth's mantle. Analysis of these fluids is possible if
they remain entrapped in the diamond during growth, but such fluid inclusions are rarely observed in gem-
quality stones. We investigated recently-discovered thin films surrounding mineral inclusions, previously
described as silicic fluid rims (SFR) containing $\text{Si}_2\text{O}(\text{OH})_6$ and $\text{Si}(\text{OH})_4$, in gem-quality lithospheric diamonds from
Siberia. Using micro-Raman spectroscopy and LA-ICPMS depth-profiling, we obtained compositional data of
the SFR surrounding both silicate and non-silicate inclusions. Slow LA-ICPMS depth-profiling at the diamond-
inclusion interface enabled differentiating the respective diamond/fluid/inclusion contributions, which
allowed detection of Ba, Sr,  b, La, Nd, and Th from the fluid rim. The trace element ratios are elevated
relative to primitive mantle ratios. The $(\text{La}/\text{Nb})_N$ $(\text{Ba}/\text{Nb})_N$ and $(\text{Nd}/\text{Nb})_N$ $(\text{Th}/\text{Nb})_N$ systematics of the fluid align
with those of high-density fluids (HDFs) in *fibrous* diamonds, suggesting a high-Mg carbonatitic composition.
Despite differences in the density of the SFR and HDFs, our results demonstrate that both have similar trace
element compositions. Considering the ubiquitous occurrence of silicic fluids at mineral inclusion rims, our
geochemical evidence reveals that gem and *fibrous* diamonds share a common mantle fluid origin.

Introduction

[revised manuscript text omitted]

**Results and Discussion**

**Silicic fluids around mineral inclusions and their origin**

We investigated seven gem-quality diamonds from Siberia. Deconvolution of FTIR spectra using the
 DiaMap_v18^{19,20} spreadsheet revealed three *Type IaA* diamonds (Int3-4, UD3-1, UD3-2), and three *Type IaB*
 diamonds (A3-4, A3-1, 7994; Table 1), one diamond (SOB9) was not classified.

**Table 1 Spectroscopic results for fluid and mineral inclusions in diamond.** Compilation of FTIR, Raman single spot, Raman mapping,
 and depth profiling results, together with the provenance for each investigated fluid and mineral inclusion from one diamond sample.
 The paragenetic type was assessed from the typical assemblage for each mineral inclusion^{6,21}. Raman intensities for fluid film signal
 (f1 lies between 646 cm⁻¹ and 677 cm⁻¹; f2 lies between 760 cm⁻¹ and 808 cm⁻¹) are background corrected for all the inclusions²², except
 for A3-4, SOB9, and Int3-4_1.

Provenance	***Dia. type	Mineral inclusion	Paragenetic type	LA-ICPMS	Raman shift ° f1 [cm ⁻¹]	Intensity 1	Raman shift °° f2 [cm ⁻¹]	Intensity 2	Intensity ratio (f1/f2)	Fluid distribution
Aikhal										
A3-4	1aB	omph. cpx	eclogitic	**x	664	490	808	335	1.5	patchy
A3-1	1aB	mgchr	peridotitic	-	673	14637	803	9647	1.5	enveloping
Internatsionalnaya										
Int 3-4_1	1aA	mgchr	peridotitic	**x	677	15235	808	12029	1.3	enveloping
Int 3-4_2	1aA	mgchr	peridotitic	-	668	3921	799	2560	1.5	unknown
Udachnaya										
UD3-1_1	1aA	olivine	peridotitic	**x	671	876	803	566	1.5	patchy
UD3-1_2	1aA	olivine	peridotitic	**x	664	1877	760	1160	1.6	patchy
UD3-1_3	1aA	olivine	peridotitic	n.m.	660	1379	799	1246	1.1	enveloping
UD3-1_4	1aA	olivine	peridotitic	**x	668	9760	795	7761	1.3	enveloping
UD3-1_5	1aA	olivine	peridotitic	n.m.	n.m.	n.m.	n.m.	n.m.	n.m.	enveloping
UD3-2_1	1aA	olivine	peridotitic	**x	660	1930	790	1211	1.6	enveloping
UD3-2_2	1aA	olivine	peridotitic	**x	646	1884	777	1098	1.7	unknown
Siberia (*u.p.)										
7994	1aB	olivine	peridotitic	**x	660	3111	782	2169	1.4	enveloping
SOB9	n.m.	mgchr	peridotitic	n.m.	668	39822	799	27649	1.4	patchy

Notes:

[revised manuscript text omitted]

Signal due to inclusions generally appears after >30 minutes of ablation (depths >100 μm, Supplementary
Information 1) for *mineral indicative* depth profiles, and earlier for *fluid indicative* profiles (<21 min; depths
<100 μm). This suggests that signal due to the fluid rim around shallower inclusions is easier to detect than
from deeper inclusions. Signal of deeper inclusions may be drowned out due to downhole fractionation and
pit tailing or diamond graphitisation during ablation (Extended Fig. 10b, c, e, f; for pit tailing)^{36,37}.

To evaluate the chemical composition of the SFRs, data were internally standardised to mean global Si contents
for the individual mineral inclusions (methods). The fluid-related intervals in the depth profiles of 7994 and
UD3-1_4 were extracted and plotted in $(La/Nb)_N$ - $(Ba/Nb)_N$ and $(Nd/Nb)_N$ - $(Th/Nb)_N$ space (Fig.2b,d; Extended
Fig. 9). Fluid-related trace element ratios, from peaks, above and below the olivine inclusions, cluster together
(Fig. 2b,d; Extended Fig. 9), indicating a fluid phase with a consistent composition. They fall near or above
primitive mantle values and align with high-density fluids previously reported in *fibrous* diamonds
The detection-limit (LoD) filtered mean SFR ratios from the fluid related interval in 7994 fall within the high-
 188 Mg carbonatitic field, while UD3-1_4 shows a similar trend but partly overlaps with the saline field due to lower
 Th and Nd abundances (Extended Fig. 9). Despite this variation, and the trend of data from low-Mg carbonatitic
 (LMC)-silicic compositional field overlapping the saline field, the SFRs surrounding olivine inclusions in 7994
 and UD3-1_4 are best interpreted as high-Mg carbonatitic, consistent with the global systematics of HDFs
 observed in *fibrous* diamonds^{7,9,38-40}.

**Figure 2 Chemical depth profiles and PM-normalized systematics of SFR signals around olivine inclusions in diamond.** (a) Depth profile
 of sample UD3-1_4 showing the onset of signal due to the olivine-inclusion and SFR assemblage after ~1300s of ablation. (c) Depth
 profile of sample 7994 reaching the inclusion after ~850s. Below the depth profile, a post-ablation cross-cut image of sample 7994
 illustrates which parts of the signal correspond to diamond background versus the inclusion assemblage. For both depth profiles in (a,
 c), blue dashed lines indicate the plateau-shaped olivine signal, while the black and red dotted lines mark the fluid signal above and
 below the mineral inclusion. Olivine is indicated by Mg, Fe, Cr, and Al; peak-shaped signals of Ba, Sr, Ce, Nb, La, Nd, and Th correspond
 to the fluid. (b, d) Primitive mantle-normalised⁴¹ data extracted from the SFR-related signal of sample 7994 (b) and UD3-1_4 (d), shown
 in $(La/Nb)_N$ - $(Ba/Nb)_N$ space. Black and red dots represent non-LoD-filtered data from the first (above-inclusion) and second (below-
 inclusion) fluid intervals, corresponding to the black and red dotted lines in the depth profiles. Triangles indicate LoD-filtered means.
 Coloured fields represent different HDF compositions, typically analysed using the offline LA-method^{7,9,38-40}. Grey diamonds show
 invisible-fluid compositions in gem-quality diamonds⁵. Blue crosses in (c, d) represent HDFs in fibrous diamond analysed with a similar
 LA-ICPMS approach⁸. Black lines indicate primitive mantle ratios.

Implications of similar fluids in gem and fibrous diamonds

The ubiquitous presence of fluid rims around mineral inclusions suggests they play a fundamental role in
diamond genesis, either as a primary growth medium or as a residual metasomatic product. Fluid rims were
observed around all investigated inclusions, independent of paragenesis, mineral type, diamond type, or
sample locality (Fig. 1, Table 1). Previously, similar fluids were mostly reported in *fibrous* diamonds^{7,9,38,40,42} or
along twin planes in gem-quality diamonds^{5,10} with origins attributed to mantle melts, metasomatic reactions,
or subduction-related fluids. The optical invisibility of such fluid inclusions and related analytical difficulties
have limited previous investigations.

Depth profiles around olivine inclusions 7994 and UD3-1_4 reveal a distinct fluid signal outlining the mineral-
related plateau, consistent with 3D information from micro-Raman maps and depth profiles (Extended Fig. 3b,
4d). Trace elements (Sr, Nb, Ba, Ce, La, Nd, Th) occur exclusively in the fluid, with low Nd contents and no Th
detected around the deeper inclusion UD3-1_4, likely due to signal attenuation. While Nb is marginally
compatible in olivine, other detected trace elements are typically not abundant^{26,27}, to the point that they are
not recommended for analysis when studying olivine inclusions in diamonds²⁶. Thus, we consider a fluid-
derived origin for these elements.

The presence of Cs suggests a subducted crustal component⁴³, while Ba and Sr anomalies align with saline
HDFs attributed to a subduction-related origin⁴⁴ where the $(La/Nb)_N$, $(Ba/Nb)_N$ and $(Nd/Nb)_N$, $(Th/Nb)_N$ systematics
match HDF compositions from *fibrous* diamonds, implying similar parental fluids for the fluid rims observed in
gem-quality and *fibrous* diamonds. Major-element contents of HDF-bearing diamonds reveal a continuum
from LMC to silicic, and from saline to high-Mg carbonatitic compositions^{42,44-46}. Silicic fluid rims fall within
high-Mg carbonatitic compositions, which likely formed through interaction of saline HDFs with peridotite³⁹.
This is consistent with the Ba, Ce, and Sr enrichment as the occurrence of the SFRs around olivine, a typical
peridotitic inclusion. For SFRs surrounding eclogitic inclusions, compositions would be expected to fall within
the LMC to silicic HDF compositional field⁴⁴.

Moreover, recent observation of silicic fluids along twin planes in gem-quality diamonds¹³ suggests that
~~previously invisible~~ nano fluid inclusions⁵ likely share the same origin as the silicic fluid rims investigated in this
study. This interpretation is reinforced by our observation that the SFRs fall within the compositional range of
HDFs from *fibrous* diamonds and align with these *invisible* nano fluids (Fig.2, Extended Fig. 9).

In contrast, the discrepancy in physical properties, such as density (HDFs: high-density; SFR: low-density), and
the fact that HDFs are found as partly recrystallised inclusions, while the SFR shows no signs of recrystallisation,
suggests distinct histories. Differences in these properties may reflect varying formation conditions for both
fluid types, potentially highlighting the presence of a less fractionated fluid phase responsible for gem-quality
diamond formation, compared to a more evolved hydrous fluid responsible¹¹ for the formation of HDF-bearing
*fibrous* diamonds. Such parental fluids may have previously coexisted with a hydrous silicate melt, as suggested
by experiments⁴⁷.

The SFR is detected above and below inclusions and is preserved during initial exposure with laser ablation,
contrary to expectations for a residual pressurized fluid. Its persistence for >100 seconds of ablation further
suggests an amorphous state¹ which is similar to that of HDFs, being viscous at mantle conditions¹.

The ubiquitousness of SFRs around inclusions in diamond, combined with our results, highlights the
importance of considering it as a post-entrapment diffusion interface, enabling element fractionation between
the fluid and mineral inclusion. Fluid mobile and radiogenic elements like Th (U-Pb dating) or Sr (Rb-Sr

dating)^{17,48}, which we detect in the fluid rim, may directly affect age determinations¹⁸. Our research
demonstrates the similarity of fluids enclosed in gem-quality and fibrous diamonds, highlighting the need to
account for silicic fluid rims in future geochemical studies of inclusions in diamond.

References

[revised manuscript text omitted]

- 43. Bellot, N. *et al.* Origin of negative cerium anomalies in subduction-related volcanic samples:
Constraints from Ce and Nd isotopes. *Chem. Geol.* **500**, 46–63 (2018).
- 44. Izraeli, E. S., Harris, J. W. & Navon, O. Brine inclusions in diamonds: a new upper mantle fluid. *Earth*
*Planet. Sci. Lett.* **187**, 323–332 (2001).
- 45. Klein-BenDavid, O. *et al.* High-Mg carbonatitic microinclusions in some Yakutian diamonds—a new
type of diamond-forming fluid. *Lithos* **112**, 648–659 (2009).
- 46. Schrauder, M. & Navon, O. Hydrous and carbonatitic mantle fluids in fibrous diamonds from Jwaneng,
Botswana. *Geochim. Cosmochim. Acta* **58**, 761–771 (1994).
- 47. Bureau, H. *et al.* The growth of fibrous, cloudy and polycrystalline diamonds. *Geochim. Cosmochim.*
*Acta* **77**, 202–214 (2012).
- 48. Timmerman, S. *et al.* Sublithospheric diamond ages and the supercontinent cycle. *Nature* **623**, 752–
756 (2023).

Fluids in gem-quality and fibrous diamonds are compositionally similar

Aleksandr Rakipov^{1,2,3*}, Alan B. Woodland^{1,2}, Fabrizio Nestola³, Matilde Galie^{1,3}, Martha G. Pamato³,
Davide Novella³, Maxwell C. Day³, Wolfgang Müller^{1,2}, Tobias Erhardt 
¹*Institut für Geowissenschaften, Goethe University, 60438 Frankfurt am Main, Germany*

²*Frankfurt Isotope and Element Research Center (FIERCE), Goethe University, 60438 Frankfurt am Main, Germany*

³*Department of Geoscience, University of Padova, 35131 Padova, Italy*

**Corresponding author*

Materials, methods, and data protocol

We have investigated a total of thirteen mineral inclusions from eight gem-quality diamonds. All the diamond samples originate from the Siberian craton, including two Aikhal diamonds (A3-4 and A3-1), one Internatsionalnaya diamond (Int3-4), two Udachnaya diamonds (UD3-1 and UD3-2), and two diamonds from unknown Siberian pipes (SOB9 and 7994). Accordingly, the studied diamonds originate from at least four different kimberlitic pipes. Most of the kimberlites that host diamonds in the Siberian craton are of Palaeozoic age, emplaced between 344 and 380 Ma⁴⁹.

The samples range in size from 1 to 3 mm and contain single to multiple (up to eight) inclusions. The inclusions vary in diameter from the smallest at 87 μm (UD3-1_4) to the largest at 437 μm (SOB9). Inclusions smaller than 80 μm were excluded, as chemical depth profiling and distinguishing between fluid and mineral inclusions using LA-ICPMS becomes increasingly challenging at smaller scales. All investigated inclusions display rounded morphologies. Except for one diamond (Inclusions: UD3-1_3 and UD3-1_4), none were polished. All inclusions remained fully enclosed within the diamond host, allowing for in situ characterisation. It was verified via optical microscopy that all cracks in the diamond are sealed and do not extend to the surface.

The depths of the inclusions were measured using the optical Keyence Microscope (VHX600) at the Goethe University Frankfurt. Note that these depth estimates do not account for the optical effects of diamond and its high refractive index⁵⁰. Apparent Depths range from the shallowest at 72 μm (7994) to the deepest at 215 μm (UD3-2_1). A compilation of the inclusion depths is provided in Supplementary Information 1; corresponding images can be found in Extended Figure 1.

Fourier-Transform Infrared (FTIR) and confocal micro-Raman analysis

Fourier-transform infrared (FTIR) measurements were conducted at the University of Padova using a Thermo Fisher Nicolet iN10 Infrared Microscope equipped with a KBr beam splitter and an LN-cooled MCT detector. Spectra were collected at a resolution of 2 cm^{-1} over the range of 750-4000 cm^{-1} , averaging 64 scans with a total scan time of 22.4 seconds. Before FTIR analysis, the sample was cleaned and mounted on a BaF₂

background window. The raw spectra were baseline-corrected, normalised to a thickness of 1 cm, and the N-
region was deconvoluted using the Excel spreadsheet DiaMap_v18, applying standard procedures from Howell
et al.^{19,20}.

Raman spectra were collected using a WITec alpha 300R confocal micro-Raman Microscope at Goethe
University Frankfurt (GUF), following prior training in inclusion analysis in diamond at the University of Padova.
The measurements covered a spectral range from 0 to 3900 cm⁻¹. A 50x objective was used, along with a 532
43 nm excitation laser operating at approximately 40 mW before the objective. A holographic grating with 600
grooves/mm was applied. The instrument was calibrated using an Ar-Hg spectral lamp, and its performance
was verified prior to measurements based on the 1300 cm⁻¹ line of silicon.

For single-spot analyses, the exposure time varied between 0.1 and 0.2 seconds, with accumulations ranging
from 5 to 20 scans. Raman spectra of mineral inclusions were compared to reference compounds available in
the RamanCrystalHunter database²². Mapping and depth profiling were performed on areas ranging from 80 ×
80 μm² to 400 × 400 μm², with depth profiles extending between 50 μm and 400 μm. Step sizes ranged from
0.5 to 2 μm, and the exposure time for each measurement was set to 0.05 seconds (Supplementary Information
1).

To identify the silicic fluid rim (SFR), Raman maps and depth profiles were typically filtered based on
characteristic wavenumber centres: 840 cm⁻¹ (width: 100 cm⁻¹), 740 cm⁻¹ (width: 200 cm⁻¹), or 660 cm⁻¹ (width:
100 cm⁻¹), depending on the coexisting inclusion type. However, certain mineral inclusions exhibit major peaks
in similar wavenumber ranges as silicic fluids, which can complicate signal separation. For instance, omphacitic
clinopyroxene (around 690 cm⁻¹), olivine (around 825 cm⁻¹ and 858 cm⁻¹), and magnesiochromite (a broad peak
at around 690 cm⁻¹ with a shoulder near 800 cm⁻¹) overlap with key fluid rim signals. In cases where filtering a
mixed signal was unavoidable, mineral-related contributions to the Raman maps could not be completely
excluded, and maps and depth profiles were filtered at a position of 840 cm⁻¹ (width: 100 cm⁻¹). However, the
resulting heat maps still provide a clear visual distinction between fluid and mineral inclusions, supported by
independent identification of the fluid rim following the mapping (Fig. 1; Extended Fig. 2-5).

Spectral features of the silicic fluid rim (SFR)

The spectral features of the SFR observed in our study are consistent with those described by Nimis et al.¹².
However, recent experimental studies, which demonstrate that an amorphous silicic component does not
coexist with water under pressure-temperature conditions relevant for the lithospheric mantle⁵¹, raise the
question of whether the SFR can be considered hydrous.
In our spectra, we occasionally observe features that have previously been attributed to a hydrous component:
a weak, broad band near 1650 cm⁻¹ associated with H₂O bending and a shoulder above 3400 cm⁻¹ related to
OH stretching. While the H₂O bending at 1650 cm⁻¹ spatially correlates with the silicic component in micro-
Raman maps (Extended Fig. 5e, f), we could not establish the same relation for the OH stretching signal, which
would be required to provide definitive evidence for a hydrous component within these rims. Moreover, this
signal (Extended Fig. 5a), and the analogous signal attributed to OH-stretching by Nimis et al.¹² is much broader
(starting from ~2000 cm⁻¹) than would be expected for OH-stretching due to H₂O or Si₂O(OH)₆ or Si(OH)₄ groups.
The broad shoulder above 3400 cm⁻¹ reported by Nimis et al.¹² and confirmed here may alternatively reflect
fluorescence and an associated artifact of background correction. Furthermore, it is important to consider
potential OH-stretching from the mineral inclusions (Extended Fig. 5b; OH-signal in olivine), ensuring it is not
mistaken for the contribution from the SFR. There also appears to be no correlation between the signal
attributed to H₂O bending and OH-stretching. For example, where the signal attributed to H₂O bending is

relatively intense (Extended Fig. 5c, d), no OH-stretching signal is observed, and *vice-versa* (Extended Fig. 5a).
Despite the higher sensitivity of FTIR to (OH) compared to Raman, attempts to detect OH-stretching or H₂O
bending signal in the infrared have been unsuccessful. However, it is possible this may be due to the large
volume of diamond sampled by transmission FTIR analyses coupled with the small size of the fluid rims. Thus,
although our observations suggest the possibility of a hydrous contribution, they do not provide definitive
evidence. Consequently, more work is still required to constrain the composition and physical properties of the
fluid rim:

Chemical LA-ICPMS depth profiles

Single spot trace elemental analysis of the fairly deep-seated (70-200 μm) mineral and fluid inclusions was
performed by slow depth-profiling using laser ablation inductively coupled plasma tandem mass spectrometry
(LA-ICP-MS/MS). Specifically, we utilized a Triple Quadrupole ICP-MS/MS (Agilent 8900) coupled to a custom-
built Dual-Wavelength (157 and 193 nm) LA system, operated at 193 nm⁵² at the Frankfurt Isotope and Element
Research Centre (FIERCE) of Goethe University Frankfurt. Before measurement, the diamonds were laser-
engraved with small caret symbols, cleaned with ethanol, and imaged with a Keyence Microscope. This image
was then aligned with the live laser camera before the LA-ICP-MS/MS measurement, using the engraved
symbols as reference points visible in the live view of the laser ablation system. This procedure was used for
orientation and improved targeting of the deep-seated mineral inclusions. This was necessary because some
of the inclusions were not visible in the laser camera, and due to the large impact of the parallax effect of the
off-axis viewing setup of RESOLUTION LA system in high refractive index materials⁵³.

Ablation was performed in a He atmosphere (0.3 l/min), to which Ar was added at a flow rate between 0.94
and 1.0 l/min. To enhance sensitivity and plasma stability^{54,55}, an additional diatomic gas, N₂ or H₂, was added
for analysis of samples 7994, A3-4, Int3-4_1, UD3-1_1, UD3-1_2, UD3-2_1 (3 ml/min, N₂), and UD3-2_2 and
UD3-1_4 (5 ml/min, H₂), respectively. The aerosol and carrier gas mixture flows to the ICP-MS/MS via a squid
signal smoothing device⁵³. Ultimately, H₂ is preferred due to the better Si detectability owing to the lower
background at m/z=28;29.

Laser fluence was ~12 J/cm² and the laser pulse frequency was triggered using the set to between 1.8 and 2.5
106 Hz using the Quadlock alignment device⁵⁶ to fire the laser per MS sweep, resulting in a repetition around 2 Hz,
depending on MS setup. The LA-ICP-MS/MS system was tuned to robust plasma conditions, monitored via
oxide production rates (²⁴⁸ThO/²³²Th below 0.5%), ²³⁸U/²³²Th ~ 1, and doubly charged ion production rates
(monitored via m/z=22/44) below 0.7%.

A total of 21 elements, reduced to 17 elements by eliminating ⁸⁵Rb, ¹⁷²Yb, ²⁰⁸Pb,  ¹⁴⁷Sm, and ¹⁵³Eu on day four,
were selected for analysis. They were categorised into three groups: (1) elements to distinguish between the
diamond host, fluid, and mineral inclusions (¹³C, ²⁵Mg, ²⁷Al, ²⁹Si, ⁵²Cr); (2) elements for comparison with
previously characterised fluids in *fibrous* and gem-quality diamonds (²³Na, ²⁵Mg, ²⁷Al, ²⁹Si, ⁴³Ca, ⁵⁷Fe, ⁹³Nb, ¹³⁸Ba,
¹³⁹La, ¹⁴⁶Nd, ²³²Th); and (3) rare earth elements (REEs) and alkalis of additional interest (⁴⁸Ti, ⁸⁵Rb, ¹⁴⁰Ce, ¹⁴⁷Sm,
¹⁵³Eu, ¹⁷²Yb). Following the switch to H₂ as additional diatomic gas, ²⁸Si was retained and ²⁹Si added to the
element list.

Measurements were performed in single quad mode; one total sweep took between 0.4018 s and 0.5380 s,
depending on the dwell times used and the number of measured masses (see Supplementary Information 2).

For data quantification, we overall followed the protocol of Pettke³⁷ (except step d), who established a data
procedure routine for LA-ICPMS depth profiling of fluid inclusions in quartz:

- (a) For each analysis, distinction, and integration of background interval and signal interval count rates for the
host diamond, mineral inclusion, and surrounding fluid rims.
- (b) Subtraction of the signal contribution of the host diamond from the fluid and solid-mineral inclusion signal.
- (c) Drift correction based on the external standard measurements, calculation of element concentration ratios
relative to the external standard, and, where applicable, conversion of these ratios into absolute element
concentrations using the assumed concentration of an internal standard within the mineral inclusion,
following the formula of Longerich et al.⁵⁷. NIST610 glass served as the external calibration, with NIST612
and ST.HS-G serving as secondary standards. The accuracy of secondary-standard depth profiles (up to
1303.34 s), based on comparison between measured and published average element concentrations for
NIST612 and ST.HS-G glasses is around $\pm 10\%$ (Supplementary Information 2). Deviations from this are likely
related to heterogeneities in NIST glasses, as observed for Mg, or due to the volatility of elements such as
Pb, which makes them more sensitive to downhole fractionation⁵⁸.
- (d) For internal standardisation, we used literature-derived global mean SiO₂ contents of corresponding
mineral inclusions in diamond: olivine 41.02 wt%; omphacitic clinopyroxene 54.79 wt%; magnesiochromite
0.22 wt%. Major-element compositions of these inclusions were taken from established reference
datasets⁵⁹. The inclusions were verified by Raman spectroscopy.
- (e) The element concentrations from c) were filtered for the respective limit of detection, which was set to 3
standard deviations of the background signal. The limit was calculated using blank analysis and determined
for each pulse based on the formula from Longerich et al.⁵⁷.

This quantification procedure was tested with the Al-in-olivine thermometer for the depth profile of sample
7994, which yields a minimum temperature of 936°C (calculated for 5GPa; SD=11°C; n=261 over 950-1060 s of
the depth profile; avg. Al $\sim 11.92 \mu\text{g/g}$). This temperature is considerably low for a lithospheric diamond,
plotting in the lower boundary of the reported equilibration temperature of peridotitic inclusions from
lithospheric diamonds⁶⁰, and this condition lies just outside the experimental calibration range (1000°C to
1300°C) of the thermometer³¹. Further, it is not possible to know whether the olivine inclusion was originally
in equilibrium in a garnet peridotite assemblage. However, this lower estimate represents a reasonable
temperature for the Siberian craton geotherm³² and demonstrates that our data reduction technique is robust.

It is noteworthy that the elemental composition of the fluid rims has previously not been characterised by any
method and, therefore, cannot be directly internally standardised. Consequently, the quantification applied
here  permits the use of trace element ratios for comparison with prior studies. Supplementary Information 2
provides background-corrected, quantified data for all depth profiles in Fig. 2 and Extended Figs. 6-8, along
with details of the LA-ICP-MS/MS setup and the accuracy of secondary-standard depth profiling for each
measurement day.

**References**

- 49. Kjarsgaard, B. A. *et al.* A Review of the Geology of Global Diamond Mines and Deposits. *Rev. Mineral.*
*Geochem.* **88**, 1–117 (2022).
- 50. Zaitsev, A. M. *Optical Properties of Diamond: A Data Handbook.* (Springer Science & Business Media,
2013).
- 51. Marras, G. *et al.* High-pressure and temperature investigation of silicic acid ± water with implications
for the ice polymorphism, lunar moganite and diamonds growth media. *Prog. Earth Planet. Sci.* **12**, 78
(2025).
- 52. Erhardt, T. *et al.* Rationale, design and initial performance of a dual-wavelength (157 & 193 nm) cryo-
LA-ICP-MS/MS system. *J. Anal. At. Spectrom.* **40**, 2857–2869 (2025).
- 53. Müller, W., Shelley, M., Miller, P. & Broude, S. Initial performance metrics of a new custom-designed
ArF excimer LA-ICPMS system coupled to a two-volume laser-ablation cell. *J. Anal. At. Spectrom.* **24**, 209–
214 (2009).
- 54. Guillon, M., Heimgartner, P., Kopajtic, Z., Günther, D. & Günther-Leopold, I. A laser ablation system for
the analysis of radioactive samples using inductively coupled plasma mass spectrometry. *J. Anal. At.*
*Spectrom.* **22**, 399–402 (2007).
- 55. Durrant, S. F. Feasibility of improvement in analytical performance in laser ablation inductively coupled
plasma-mass spectrometry (LA-ICP-MS) by addition of nitrogen to the argon plasma. *Fresenius J. Anal.*
*Chem.* **349**, 768–771 (1994).
- 56. Ashley Norris, C., Danyushevsky, L., Olin, P. & R. West, N. Elimination of aliasing in LA-ICP-MS by
alignment of laser and mass spectrometer. *J. Anal. At. Spectrom.* **36**, 733–739 (2021).
- 57. Longerich, H. P., Jackson, S. E. & Günther, D. Inter-laboratory note. Laser ablation inductively coupled
plasma mass spectrometric transient signal data acquisition and analyte concentration calculation. *J. Anal.*
*At. Spectrom.* **11**, 899–904 (1996).

- 58. Evans, D. & Müller, W. Automated Extraction of a Five-Year LA-ICP-MS Trace Element Data Set of Ten
Common Glass and Carbonate Reference Materials: Long-Term Data Quality, Optimisation and Laser Cell
Homogeneity. *Geostand. Geoanalytical Res.* **42**, 159–188 (2018).
- 59. Stachel, T. Diamond Inclusion Database. Borealis <https://doi.org/10.7939/DVN/EJUE1G> (2021).
- 60. Stachel, T. & Harris, J. W. Formation of diamond in the Earth's mantle. *J. Phys. Condens. Matter* **21**,
364206 (2009).
- 61. Sun, Q. The Raman OH stretching bands of liquid water. *Vib. Spectrosc.* **51**, 213–217 (2009).
- 62. Besemer, M., Bloemenkamp, R., Ariese, F. & Manen, H.-J. van. Identification of Multiple Water–Iodide
Species in Concentrated NaI Solutions Based on the Raman Bending Vibration of Water. *J. Phys. Chem. A*
**120**, 709–714 (2016).

**Acknowledgements**

We express our gratitude to the Deutsche Forschungsgemeinschaft grant (DFG: WO 652/38-1), acquired by
190 A.B.W., for providing financial support for the LA-ICPMS analyses. This project was financially supported by
191 European Research Council (ERC) grant Discovering **M**ineral **n**Anoinclusions in **DiAM**ond (MADAM:
101199315) acquired by F.N., which is gratefully acknowledged. We are grateful for the technical support of
Alexander Schmidt and Richard Albert with microscope imaging. FIERCE is financially supported by the Wilhelm
and Else Heraeus Foundation and by the Deutsche Forschungsgemeinschaft (DFG: INST 161/921-1 FUGG, INST
161/923-1 FUGG and INST 161/1073-1 FUGG), which is gratefully acknowledged. This is FIERCE contribution
No.XXX.

*Extended Figure 1 Diamond samples and their mineral inclusions. Capital letters (A-G) show whole-sample images, while small letters*
 *(a-g) present enlarged views of inclusions from the corresponding host diamond.*

*Extended Figure 2 SFR distribution around inclusions A3-4, A3-1, INT3-4_1, INT3-4_2, UD3-1_1, and UD3-1_2. Raman map filtered for*
 *the indicative SFR signal, shown together with colour-coded depth profiles across omphacitic clinopyroxene in (a), magnesiochromite*
 *(b-d), and olivine (e) inclusions. The SFR distribution is classified as enveloping for inclusions in (b) and (c) and patchy in (a) and (e). See*
 *Figure 1 for mineral inclusion and SFR spectra.*

a) UD3-1_3

b) UD3-1_4

c) UD3-1_5

*Extended Figure 3 SFR distribution around olivine inclusions UD3-1_3, UD3-1_4, and UD3-1_5. Raman map filtered for the indicative*
*SFR signal, with depth profiles indicated in blue. The fluid distribution is classified as enveloping for all inclusions. See Figure 1 for mineral*
*inclusion and SFR spectra.*

Extended Figure 4 SFR distribution around inclusions UD3-2_1, UD3-2_2, SOB9, and 7994. Raman map filtered for the indicative SFR signal, with different depth profiles according to colour across olivine in (a-b), (d), and magnesiochromite inclusions in (c). The distribution is classified as enveloping for inclusions in (a) and (d) and as patchy in (c). See Figure 1 for mineral and SFR inclusion spectra.

Extended Figure 5 Spectral evidence for the hydrous component: (a) broad absorption previously attributed to OH-stretching by Nimis et al.¹²; (b) OH-stretching peak at 3574 cm^{-1} due to olivine of sample 7994⁶¹; (c, d) SFR signal and broad peaks at $\sim 1650 \text{ cm}^{-1}$ that may be due to H_2O -bending for inclusions SOB9 and UD3-2_1⁶². (e, f) Raman maps filtered for the $\sim 1650 \text{ cm}^{-1}$ band (H_2O -bending; in panels 1. and 2.) and the $\sim 660 \text{ cm}^{-1}$ band (SFR; in panels 3. and 4.).

Extended Figure 6 Chemical depth profile of olivine inclusion (7994). The upper panel shows the multi-element depth profile of the inclusion 7994, while the panels below display individual single-element depth profiles

Extended Figure 7 Chemical depth profile of olivine inclusion (UD3-1_4). The upper panel shows the multi-element depth profile of the inclusion UD3-1_4, while the panels below display individual single-element depth profiles.

a) A3-4

b) UD3-2_2

c) Int3-4_1

d) UD3-1_2

e) UD3-1_1

f) UD3-2_1

*Extended Figure 8 Chemical depth profiles of inclusions A3-4, UD3-2_2, Int3-4_1, UD3-1_2, UD3-1_1, and UD3-2_1. Omphacitic*
 *clinopyroxene (a) and the olivine inclusion (b) show group 2 depth profiles, in which the mineral inclusion and the SFR cannot be resolved,*
 *likely due to a patchy SFR distribution or the high trace-element background of the omphacitic clinopyroxene. The magnesiochromite*
 *inclusion (c) and the olivine inclusions in (d-f) display group 1 depth profiles, in which only major-element (Mg, Al, Ti, Cr, and Fe)*
 *signals were detected, and do not allow separation of the mineral inclusion from the SFR.*

*Extended Figure 9 Primitive mantle-normalised data extracted from the SFR-related signal of sample 7994 (a) and UD3-1_4 (b),*
 *shown in $(Nd/Nb)_N$ - $(Th/Nb)_N$ space. Black and red dots represent non-LOD-filtered data from the first (above olivine inclusion) and*
 *second (below-inclusion) fluid intervals, corresponding to the black and red dotted lines in the depth profiles (Fig.1). Triangles indicate*
 *LoD-filtered means. Coloured fields represent different HDF compositions, typically analysed using offline LA-ICPMS (cf. legend)^{7,9,38-40}.*
 *Grey diamonds show invisible-fluid compositions in gem-quality diamonds⁵. Blue crosses represent HDFs in fibrous diamond analysed*
 *with an online LA-ICPMS approach⁸. Black lines indicate primitive mantle ratios.*

*Extended Figure 10 Laser ablation pit images of diamond samples. Post laser ablation (LA) side-view images of diamond samples*
 *showing the ablation pits. Images (A) and (D-F) illustrate successfully targeted measurements, where the LA reached the inclusions,*
 *whereas (B) and (C) show pits that thinned out before the inclusions were reached by ablation.*